evolution, genetics, developmental biology

water strider, polyphenism, insulin receptor, RNA interference, developmental plasticity, FOXO

**Authors for correspondence:**
Erik Gudmunds
e-mail: erik.gudmunds@ebc.uu.se
Arild Husby
e-mail: arild.husby@ebc.uu.se

# Photoperiod controls wing polyphenism in a water strider independently of insulin receptor signalling

Erik Gudmunds[1], Shrinath Narayanan[1], Elise Lachivier[1], Marion Duchemin[1], Abderrahman Khila[1,2] and Arild Husby[1]

[1]Evolutionary Biology, Department of Ecology and Genetics, Uppsala University, Norbyvägen 18D, 75236 Uppsala, Sweden
[2]Institut de Génomique Fonctionnelle de Lyon, Université de Lyon, Université Claude Bernard Lyon, CNRS UMR 5242, Ecole Normale Supérieure de Lyon, 46, allée d'Italie, 69364 Lyon Cedex 07, France

EG, 0000-0001-9496-8976; AK, 0000-0003-0908-483X; AH, 0000-0003-1911-8351

Insect wing polyphenism has evolved as an adaptation to changing environments and a growing body of research suggests that the nutrient-sensing insulin receptor signalling pathway is a hot spot for the evolution of polyphenisms, as it provides a direct link between growth and available nutrients in the environment. However, little is known about the potential role of insulin receptor signalling in polyphenisms which are controlled by seasonal variation in photoperiod. Here, we demonstrate that wing length polyphenism in the water strider *Gerris buenoi* is determined by photoperiod and nymphal density, but is not directly affected by nutrient availability. Exposure to a long-day photoperiod is highly inducive of the short-winged morph whereas high nymphal densities moderately promote the development of long wings. Using RNA interference we demonstrate that, unlike in several other species where wing polyphenism is controlled by nutrition, there is no detectable role of insulin receptor signalling in wing morph induction. Our results indicate that the multitude of possible cues that trigger wing polyphenism can be mediated through different genetic pathways and that there are multiple genetic origins to wing polyphenism in insects.

## 1. Introduction

Life in changing environments where fitness optima may change rapidly [1] may lead to evolution of phenotypic plasticity when environmental cues can be used to predict future fitness optima [2]. In some cases, plasticity has evolved to induce dramatically different phenotypes such as seen in different insect polyphenisms/polymorphisms [3]. One such example is wing length polyphenism, which has emerged as a prominent model system to understand the interplay between genes and environment in causing phenotypic variation (see [4] for a recent review).

Although wing polymorphism is widespread among insects [5], most knowledge about the genetic and environmental determinants of wing morphs comes from a few species, including crickets [6], water striders [7,8], aphids [9] and planthoppers [10]. Within and across these species, there is variation in the environmental cues that determine wing morph development and the genetic mechanisms behind this variation has recently attracted interest. Functional and *in silico* analyses suggest that the insulin/insulin-like receptor signalling (IIS) is a conserved signal transduction pathway in the regulation of wing length polyphenism in several species [11]. In the brown planthopper, for example, the activity of two InRs antagonistically regulates morph determination, partly in relation to growth status of its host plant [12]. Furthermore,

knockdown of the main output transcription factor of the IIS pathway, forkhead box O (FOXO) dramatically shifts the wing morph ratio in favour of the LW morph [13,14]. Knockdown of FOXO also alters wing morph frequency in the soapberry bug *Jadera haematoloma* [15], but this effect is in the opposite direction, i.e. towards an increased frequency of the SW morph. In the fire bug, *Pyrrhocoris apterus*, knockdown of InR1 had no effect on morph frequency, whereas knockdown of InR2 and a lineage-specific intron-less copy of InR1 (called InR1a) increased the proportion of LW individuals significantly [11].

Thus, evidence from several species points to a conserved role for the IIS pathway in wing polyphenism but also shows that different species use different components within the IIS pathway and even use the same components in different ways (i.e. FOXO promotes LW in *J. haematoloma* but SW in *N. lugens*). In both planthoppers and soapberry bugs, the environmental signal producing wing polyphenism is nutrition [12,15], but it is not known whether the role of the IIS pathway is conserved in species where wing polyphenism depends on other environmental cues.

Water striders are a group of semi-aquatic insects which are highly variable in dispersal capability, mainly through variation in wing polymorphism [7,16,17]. Species may be monomorphic for long wings or short wings or may be apterous, while others display seasonal wing polyphenism or have genetic polymorphisms [18,19]. Thus, while in some species wing length does not respond to environmental cues, other species demonstrate environmentally induced wing forms [8,19,20], and different populations of these species may vary in wing morph frequency in latitudinal or altitudinal clines [7,16,21]. The environmental factors influencing wing morph determination have been investigated in a number of water striders and include photoperiod (either constant or gradually changing), temperature and nymphal rearing densities [16,19,22–26].

The variation in the environmental and genetic factors influencing wing polymorphism across the phylogeny of water striders makes them a promising model system for studies on the evolution of phenotypic plasticity [8]. However, no study to date has investigated which genes/pathways underlie wing morph induction in this group. Our aim here was to first determine the environmental factors that cause variation in wing length in the species *Gerris buenoi* (figure 1*a,b*) and second to examine if the IIS pathway is involved in wing morph determination in this species. The IIS pathway is a promising candidate given the results of studies on brown planthoppers and soapberry bugs, but also because *G. buenoi* have a lineage-specific intron-less paralogue of *inr1* [27], similar to, but independent of the intron-less *InR1a* gene in *P. apterus* (see above).

## 2. Material and methods

### (a) Gerrid cultures and rearing
The *G. buenoi* population was originally collected from a pond in Toronto, Ontario, Canada, during 2012 and was replenished in 2015 with 30 individuals from a population in the same area. The photoperiods for the stock population were from 2012 to May 2019 16 h light : 8 h dark (16 L : 8 D) and since then 22 L : 2 D. From May 2019 until present, the generation of adults in the breeding stock was primarily from individuals that had

been reared during their nymphal stages in either 12 L : 12 D or 18 L : 6 D, constituting a mix of short-winged (henceforth micropterous) and long-winged (henceforth macropterous) morphs. A smaller proportion of breeders came from nymphs reared at high densities in 22 L : 2 D which produces a small frequency of macropterous individuals.

Gerrids in the stock population were fed five times a week with frozen crickets (instar 2–3 *Acheta domestica*) and kept at room temperature. Pieces of Styrofoam were provided as a substrate for egg laying and resting.

### (b) Determination of environmental factors governing wing morph determination in *Gerris buenoi*

#### (i) Constant photoperiod experiments
Eggs were collected from the stock culture, in which adults had a mix of wing morphs or from crosses of each wing morph (only in the case of the 15 L : 9 D photoperiod) and randomly distributed into climate-controlled rooms held at different photoperiods (12 L : 12 D, 14 L : 10 D, 15 L : 9 D, 16 L : 8 D and 18 L : 6 D) with approximately 80 µEinstein (about 9400 lux) light intensity conditions, at 25°C constant temperature. These eggs were allowed to hatch in a specific box and instar 1 individuals were transferred into 38 × 19 cm experimental boxes approximately 48 h after hatching in groups of 15 individuals. Crickets (instar 2–3) were provided at least five times per week during the course of the experiments, in an amount considered *ad libitum*. Individuals that died were not replaced. Wing morph was scored at the adult stage and we defined a micropterous as an individual with forewings not stretching beyond the first abdominal segment and a macropterous with forewings stretching at least to the sixth abdominal segment (see electronic supplementary material, figure S1 for visualization of segments). We term intermediate wing sizes as mesopterous, defined by forewing length being between the first and sixth segments (7). In relation to the photoperiod conditions in the area of origin for the *G. buenoi* cultures, the 14 L : 10 D, 15 L : 9 D and 16 L : 8 D photoperiods are in the range that developing nymphs are exposed to (Toronto solstice daylength is 15 : 27 hh : mm, 16 : 39 including civil twilight).

#### (ii) Stage-specific shifts in photoperiod
To investigate the process of wing morph determination in *G. buenoi*, we performed an experiment in which we exposed nymphs to one starting photoperiod and then, at specific stages during development from instar 3–5 we shifted them (without gradual change) to a different photoperiod. A similar experimental design has been used in studies of *Aquarius paludum* [28]. We chose 12 L : 12 D and 18 L : 6 D as photoperiods, as these were highly predictive for wing morph when nymphs were exposed throughout their entire period of development (the experiments described above). We shifted individuals at six different developmental time-points as defined in the electronic supplementary material, table S1. Specifically, we collected eggs from the stock culture and hatched them in either 12 L : 12 D or 18 L : 6 D at 25°C constant temperature. Individuals in instar 1 were transferred to and held in separate boxes (38 × 19 cm) in groups of approximately 50 individuals until instar 2 when they were placed individually in plastic cups of 7 cm in diameter. Each individual nymph was fed one cricket (instar 2–3) per day throughout the experiment.

Before the subsequent shift in photoperiod, each individual was monitored for moulting to provide a record of the age of each individual within each instar. We started 45 individuals per developmental stage and shift direction; final sample sizes are given in figure 1*e,f*. To evaluate differences in wing morph determination between the two shifts in photoperiod (12 L : 12 D to 18 L : 6 D or 18 L : 6 D to 12 L : 12 D), we classified the emerging

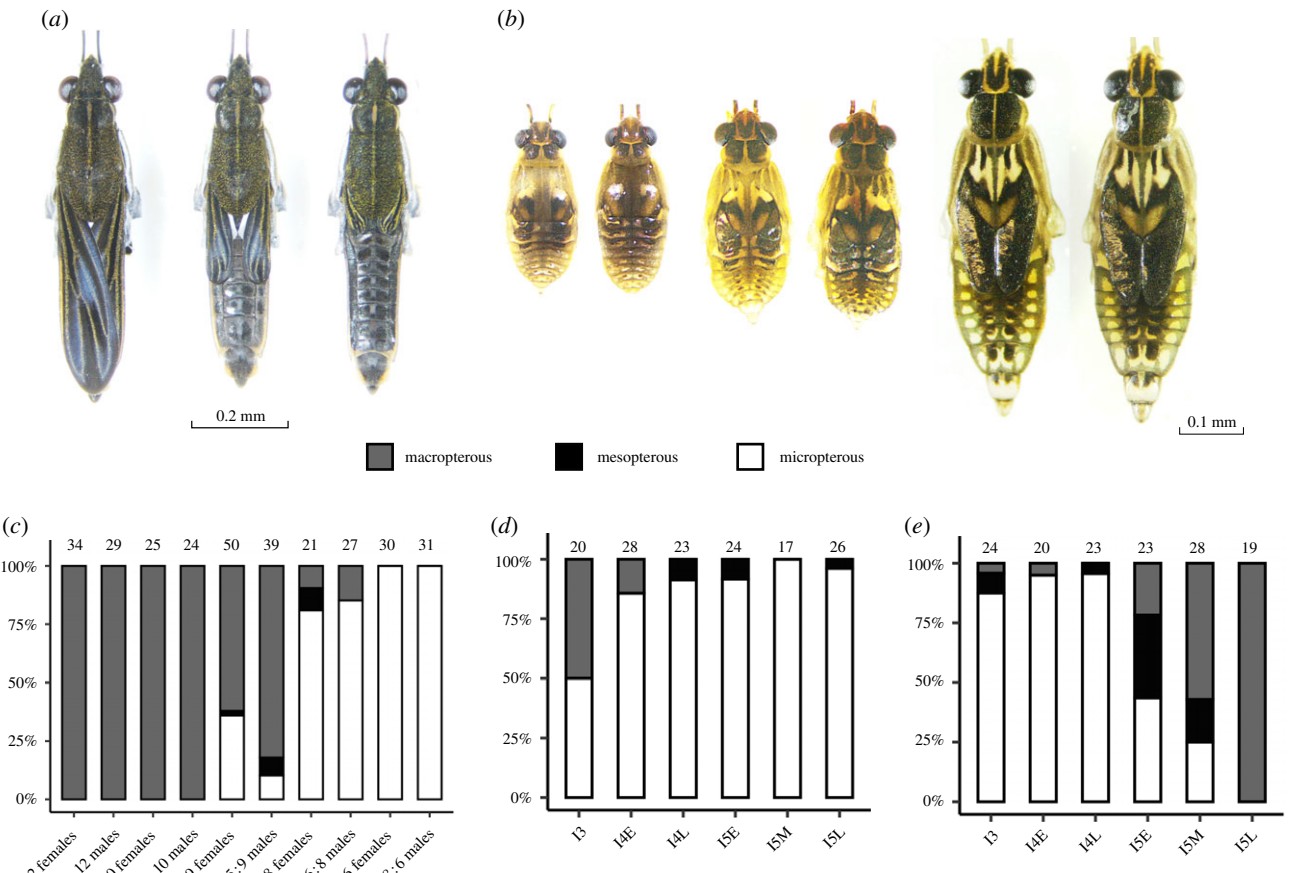

**Figure 1.** (a) *G. buenoi* long-winged (macropterous), intermediate-winged (mesopterous) and short-winged (micropterous) males, from left to right. (b) Progression of development of wing progenitor tissue (wing buds) in instar 3, 4 and 5 in 12 L : 12 D and 18 L : 6 D. Left individuals are from 12 L : 12 D and right from 18 L : 6 D. (c) Wing morph percentages of individuals raised in the indicated constant photoperiods. Sample sizes are indicated above each bar. (d,e) Wing morph percentages among individuals which experienced an 18 L : 6 D to 12 L : 12 D (d) or 12 L : 12 D to 18 L : 6 D shift (e) in photoperiod at the indicated developmental stages. Sample sizes are indicated above each bar. Abbreviations are I3 (instar 3), I4E (instar 4 early), I4L (instar 4 late), I5E (instar 5 early), I5M (instar 5 mid) and I5L (instar 5 late). See Materials and methods for specific definitions of these stages.

adult individuals into two groups which we hypothesize corresponds to two different states in polyphenic development. These states were 'committed' and 'uncommitted', defined respectively as showing the same wing morph as nymphs reared completely under the initial photoperiod, or showing the wing morph associated with nymphs reared entirely at the second photoperiod the second photoperiod. For example, an individual reared in 12 L : 12 D for early development and then shifted to 18 L : 6 D showing the micropterous morph would be classified as uncommitted at the time of photoperiod change, whereas if it showed the macropterous morph it would have been committed, with regard to the adult wing morphology. We hypothesized that at some point in development, a point of irreversibility in wing morph determination is reached, which would manifest itself in an incapability to respond to a new, otherwise highly inductive, photoperiod. A caveat with this shifting experiment is that the change in photoperiod (from 12 L : 12 D to 18 L : 6 D or vice versa, resulting in a 6 h shift) is artificial and thus our results might not reflect the same dynamics of wing morph induction that occurs in natural conditions. Furthermore, these two photoperiods are outside the range of photoperiods that any nymph from the native Toronto *G. buenoi* population would experience during nymphal development under field conditions.

We compared the frequencies of uncommitted and committed individuals between the groups using a generalized linear model (binomial error structure and logit link function), using the direction of photoperiodic shift and stage as fixed factors. Here, mesopterous individuals were grouped together with micropterous individuals to make the response variable binomial

(macropterous or non-macropterous) but the conclusions do not change if mesopterous individuals are excluded.

### (iii) Density experiments

Eggs from the stock culture were randomly distributed into rooms with photoperiods of 12 L : 12 D or 18 L : 6 D and 25°C constant temperature. Eggs were hatched in a 38 × 19 cm box and instar 1 individuals were transferred to separate growth boxes kept within each photoperiod (also 38 × 19 cm) each day for 3 days in total. The density in both the hatching boxes and the growth boxes was very high and exceeded that of the extreme density treatment (see below). After being kept in the growth boxes for approximately 4 days after hatching, instar 2 individuals were randomly selected to start the treatments. These were started by placing groups of 15, 35 or 60 individuals in 38 × 19 cm boxes, or 80 individuals in 19 × 26 cm boxes. These numbers correspond, respectively, to starting densities of 2.1, 4.8, 8.3 and 16.2 individuals per 100 cm$^2$. In the following discussion, we refer to these rearing densities, respectively, as low, medium, high and extreme. We note that the extreme density treatment is comparable to the others only in terms of relative density. Four replicates of each density treatment were established under each photoperiod and nymphs in each replicate were fed every day with crickets (instar 2–3) *ad libitum*. No replacement of dead individuals occurred. The mean relative densities at the end of the experiment were 1.6, 3.4, 5.5 and 9.4 individuals per 100 cm$^2$ for 18 L : 6 D and 1.9, 3.4 and 6.2 11.4 individuals per 100 cm$^2$ for 12 L : 12 D for the low, medium, high and extreme

Proc. R. Soc. B 289: 20212764

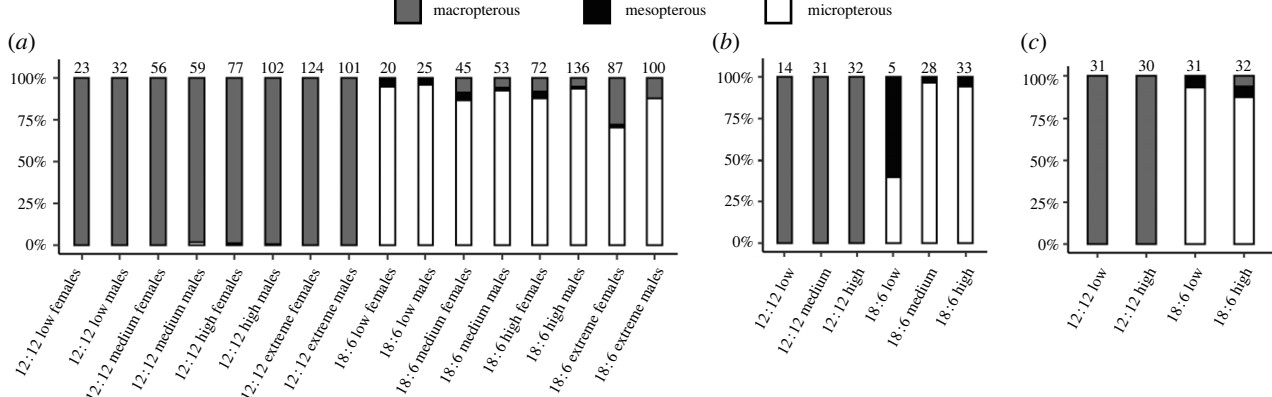

**Figure 2.** (*a*) The effect of nymphal rearing density on wing morph induction. Nymphs were reared in a range of densities (low, medium, high and extreme, 2.1, 4.8, 8.3 and 16.2 individuals per 100 cm², respectively) from instar 2 until adulthood. (*b*) Wing morph percentages of individuals raised from day 1 of instar 1 in either 12 L : 12 D or 18 L : 6 D constant photoperiods under different diet regimes. The regimes were low, medium and high, corresponding to 3, 5 or 7 crickets per week (one per day). (*c*) Wing morph percentages of individuals raised from day 1 of instar 4 in either 12 L : 12 D or 18 L : 6 D constant photoperiods under different diet regimes. The regimes were low and high, corresponding to 3 and 7 crickets per week, respectively (one per day). The results from the starvation regime (see Material and methods) are not shown as very few individuals survived until adulthood.

density treatments, respectively. The effect of starting density was tested with a generalized linear model with logit link function, where mesopterous and micropterous were pooled, so that the binomial response variable was macropterous or non-macropterous.

### (iv) Nutrition

To explore potential nutritional effects on wing morph determination, we exposed nymphs to different nutritional regimes in either 12 L : 12 D or 18 L : 6 D at 25°C constant temperature in two separate nutrition experiments. We started the first experiment with instar 1 individuals hatched in either 12 L : 12 D or 18 L : 6 D and reared these to adults under different nutritional regimes (defined below) in isolation in plastic cups of 7 cm in diameter. In the second experiment, groups of approximately 35 individuals hatched in either 12 L : 12 D or 18 L : 6 D were reared in 35 × 19 cm boxes and fed five times a week until the first day of instar 4, when all individuals were transferred to individual cups (7 cm diameter) in which they were exposed to different nutritional regimes. For each individual in the experiment, photoperiod was kept the same as hatching photoperiod (12 L : 12 D or 18 L : 6 D) throughout development. The nutritional regimes were 1, 3, 5 or 7 crickets per week (one instar 2–3 cricket per day), henceforth, we refer to these as starvation, low, medium and high, respectively. In the second experiment, only starvation, low and high were tested. In all feeding regimes crickets were removed 1 day after they were supplied. The low-nutrient regime provided a maximum 2 days without a cricket, and the medium regime provided a maximum of 1 day without a cricket. For the starvation regime, crickets were provided and removed once per week. In both experiments, moulting events and mortality was monitored each day and wing morph was scored in adults as described above. We assessed the degree of stress of the regimes by comparing developmental duration, mortality and adult body size. Each replicate was started with 36 individuals, final sample size was decreased by mortality (specific numbers are given in figure 2*b*).

### (c) RNA interference

dsRNA was synthesized from PCR products generated from genomic DNA or cDNA using the primers in the electronic supplementary material, table S2. The primers have the T7 RNA polymerase promotor sequence as overhangs (bold letters in sequences electronic supplementary material, table S2) to

facilitate generation of dsRNA through *in vitro* transcription. Each DNA template was verified with Sanger sequencing (Eurofins Genomics Mix2Seq) before *in vitro* transcription overnight with T7 RNA polymerase (Thermo Fisher Scientific). The dsRNA was treated with DNase I (Thermo Fisher Scientific) and then purified using phenol : chloroform salt extraction or GeneJET RNA Cleanup and Concentration Micro Kit (Thermo Fisher Scientific). Elution or resuspension was done in Spradling injection buffer [29]. Microinjections of dsRNA were performed with Eppendorf Celltram Vario with attached 1 mm outer diameter glass capillary needles (Narishige).

For injections, nymphs of instar 3, 4 or 5 which were hatched and reared in either 12 L : 12 D or 18 L : 6 D were sedated with $CO_2$ and gently attached on double-sided adhesive tape to hold them still. Injections were made in between the 6th and 5th abdominal segment ventrally delivering dsRNA solution of 0.2–0.3 µl for instar 3, 0.3–0.4 µl for instar 4 and 0.4–0.6 µl for instar 5, with the concentrations specified in the electronic supplementary material, table S3. After injection, replicate groups of 7–8 injected nymphs were held in either photoperiod 19 × 26 cm boxes of (1.6 individuals per 100 cm²) and fed with crickets (instar 2–3) five times a week until being phenotyped as adults (final sample size indicated in figure 3*a–c*). To confirm knockdown, we extracted total RNA using Trizol (Qiagen) from seven whole individuals (males) per dsRNA treatment 2 days after moulting into instar 4 (about 5 days after injection) and performed quantitative reverse transcriptase PCR (qRT-PCR), see electronic supplementary material, table S2 for primers. For confirmation of the effect of FOXO RNAi in wing buds, striders were injected in instar 3 and sampled 2 days after moulting into the 5th instar. Furthermore, to confirm the functionality of our RNAi protocol, we knocked down the expression of *Distalless (Dll)* with RNAi, where we injected *Dll* dsRNA in instar 3 nymphs hatched and reared in 12 L : 12 D and evaluated developmental effects expected from manipulating the expression of *Dll* (e.g. abnormal leg shape and shorter than normal wing buds). The RNAi knockdown treatment for each gene is referred to as dsGFP, dsINR1, dsINR2, dsINR1-like, dsFOXO, INR-cocktail or dsDLL below. The negative control in the RNAi experiments is the dsGFP treatment, where dsRNA with the sequence of green fluorescent protein (GFP) is injected into individuals but should not interfere with gene expression, thus controlling for potential effects of the injection procedure in itself on wing morph determination.

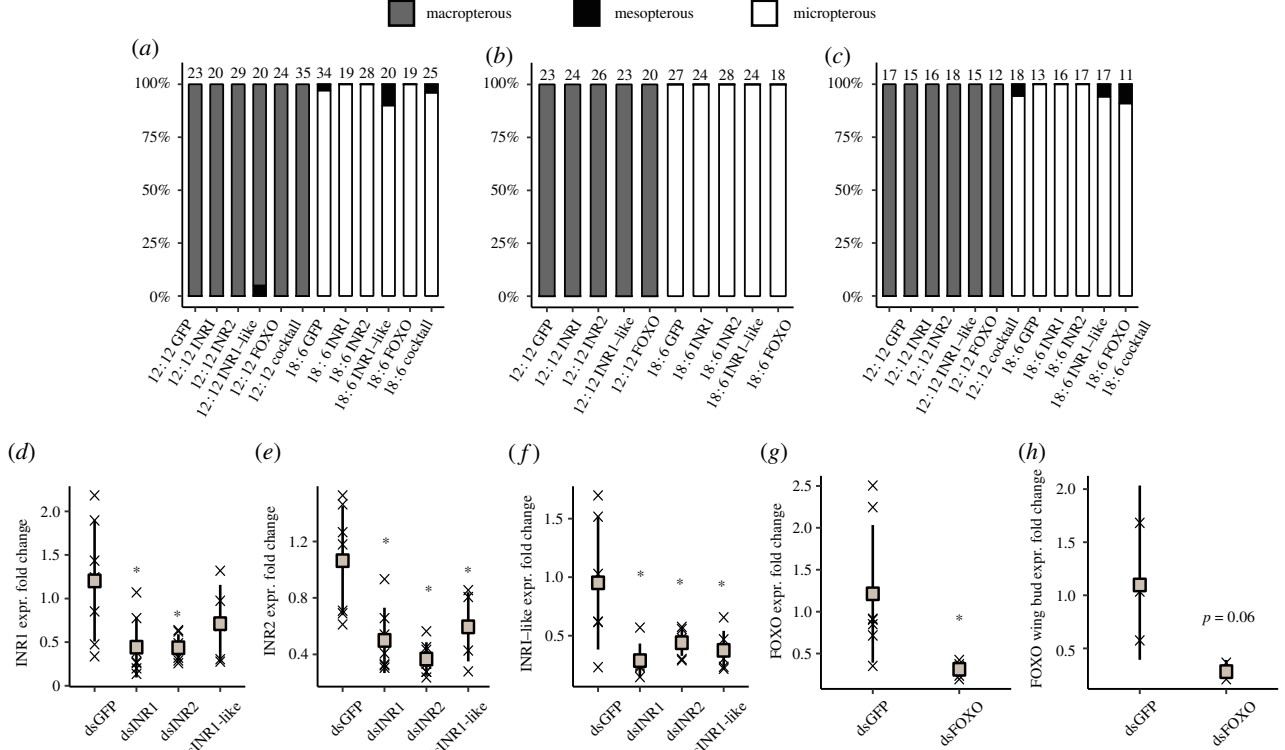

**Figure 3.** RNAi against IIS components does not alter wing morph. Individuals were injected in instar 3 (*a*), 4 (*b*), or 5 (*c*) with dsRNAs targeting either INR1, INR2, INR1-like, FOXO or all INRs (cocktail) mRNAs. GFP dsRNA was used as a negative control. Knockdown validation was performed with RT-qPCR for each gene as indicated in (*d–h*); (*h*) shows knockdown levels of FOXO mRNA in wing buds at early instar 5. Asterisks indicate a significant ( $p < 0.05$, ANOVA with Tukeys post hoc) reduction in expression compared to the dsGFP control. For (*h*) a one-sided *t*-test was performed with alternative hypothesis that FOXO expression is lower in dsFOXO compared to dsGFP treatment.

## (d) Body size measurements and data statistical analysis

Individuals used for measurements were stored in 70% ethanol at 4°C until photographed on millimetre paper using a Nikon SMZ800 stereo microscope with Nikon DS Fi1 camera attachment. Measurements on these photographed specimens were made with ImageJ (v. 1.53 k, see electronic supplementary material, figure S2 for definition of measurements). All statistical analysis was performed in R (v. 4.0.3). ANOVAs, *t*-tests, Pearson's $\chi^2$ and generalized linear models were performed within the R set of core functions (stats 4.0.3). All code and data are available in the electronic supplementary material. The estimation of pairwise sequence identity of the *inr* genes coding sequences was done with Muscle (v. 3.8.425 (alignment in Geneious (v. 2021.1.1) using standard parameters.

## 3. Results

### (a) Wing morph in *Gerris buenoi* is determined by photoperiod and is independent of parental phenotype

All nymphs became macropterous in the 12 L : 12 D and 14 L : 10 D photoperiod treatments, whereas all nymphs in 18 L : 6 D became micropterous (figure 1*c*). Intermediate frequencies of adult wing morphs resulted from the 15 L : 9 D and 16 L : 8 D photoperiod treatments (figure 1*c*). To assess whether the state of wing morph in parents influences the determination of wing morph in the offspring, we tested whether the frequencies of wing morphs in 15 L : 9 D differed between offspring generated from different controlled

crosses but no significant difference was found (electronic supplementary material, table S4, $\chi^2 = 4.66$, d.f. = 6, $p = 0.58$). We detected a significant sex-difference in wing morph frequencies in 15 L : 9 D (electronic supplementary material, table S4, $\chi^2 = 11.82$, d.f. = 2, $p = 0.002$) where females were less likely to become macropterous. Together, these results show that constant photoperiod strongly influences wing morph determination in *G. buenoi* and that it does so independently of the parental wing morph phenotype.

### (b) Exposure to 18 L : 6 D as early as instar 3 biases development toward the micropterous morph

Adult individuals differed significantly in wing morph depending on the direction of shift in photoperiod experienced as nymphs in instars 3–5 (figure 1*d,e*). For individuals experiencing the 18 L : 6 D to 12 L : 12 D shift, a tendency to develop to micropterous adults was seen at instar 3 (approx. 50%) and no macropterous individuals were found when the shift occurred after early instar 4 (i.e. these individuals were committed to micropterous development when the shift in photoperiod occurred). By contrast, individuals experiencing the 12 L : 12 D to 18 L : 6 D shift in photoperiod were seemingly uncommitted with regard to developing a particular wing morph until as late as day 2 of instar 5. The effects of instar, direction of photoperiod shift and their interaction were highly significant ($\chi^2 = 63,27$, d.f. = 5, $p = 2.56 \times 10^{-12}$, $\chi^2 = 125,87$, d.f. = 1, $p < 2.2 \times 10^{-16}$, d.f. = 5, $\chi^2 = 14,18$, $p = 0.015$, respectively). Thus, the onset of commitment to a particular wing morph differed markedly during development in either 12 L : 12 D or 18 L : 6 D. We found no

evidence of sex-differences in morph determination (electronic supplementary material, figure S3, $\chi^2 = 1.39$, d.f. = 1, $p = 0.24$ and $\chi^2 = 0.36$, d.f. = 1, $p = 0.55$ for increase or decrease shifts respectively).

### (c) Wing polyphenism in Gerris buenoi is sensitive to nymphal density but not nutrient availability

In addition to photoperiod, we also explored nymphal density and nutrition as environmental factors for wing morph determination in *G. buenoi* (figure 2). Almost all nymphs reared in 12 L : 12 D across all density regimes became macropterous (three out of 574 in total became micropterous), whereas in 18 L : 6 D most individuals became micropterous across all densities (figure 2*a*). The frequency of the macropterous morph in 18 L : 6 D significantly increased with density ($\chi^2 = 24{,}39$, d.f. = 3, $p = 2 \times 10^{-5}$). Interestingly, the frequency of macropterous individuals in 18 L : 6 D in the extreme density was higher among females than males ($\chi^2 = 18{,}23$, d.f. = 6, $p = 0.006$).

When exploring the effect of nutrient availability, the ratio of wing morphs remained unchanged in 12 L : 12 D (100% macropterous individuals), and in 18 L : 6 D only two macropterous individuals appeared of 129 in total across all dietary regimes. While we found a significant difference in the frequency of mesopterous adults among individuals treated with different nutrient regimes from instar 1 (figure 2*b*, $\chi^2 = 17{,}08$, d.f. = 2, $p = 0.0002$), sample size was very low in the low diet and thus this result should be treated with caution. Among the individuals reared in the nutrient regimes from instar 4 group (figure 2*c*), we found that the morph frequencies did not differ significantly between low- and high-nutrient regimes ($\chi^2 = 2{,}00$, d.f. = 2, $p = 0.37$), although they differed profoundly between photoperiods. We assessed how the nutrient regimes affected mortality, developmental duration and adult body size, in order to test if the restricted diet regimes were stressful. We found a clear effect of nutritional regime on mortality and developmental duration and, furthermore, adults that had been reared in restricted diets were significantly smaller than adults reared in the highest diet (electronic supplementary material, figure S4A–G). These results show that the dietary regimes that we tested were stressful for the gerrids and that wing morph determination is robust to restricted nutritional availability.

### (d) Depletion of genes in the insulin/insulin-like receptor signalling pathway does not alter wing morph

To test whether the IIS pathway has a role in controlling wing polyphenism in *G. buenoi*, we injected male and female nymphs grown in 12 L : 12 D and 18 L : 6 D during the 3rd, 4th or 5th instar with dsRNA targeting INR1, INR2, INR1-like or FOXO, or a cocktail of all three insulin receptor genes. GFP dsRNA was used as a negative control. We found no differences compared to control treatment in frequencies of the wing morphs for any of the genes subjected to gene expression knockdown in the RNAi treatments (figure 3*a–c*). A proportion (16%) of individuals in the instar 4 injection group expressed an aberrant wing phenotype (electronic supplementary material, figure S5) where one or both of the wings had abnormal morphology. This

was, however, shown to be a result of handling during injections and not from knock down of any specific gene as it was present in non-injected control individuals as well in the dsGFP negative control treatment.

We used reverse transcription-quantitative PCR (RT-qPCR) to confirm knockdown of each mRNA. All had reduced levels as compared to the dsGFP treatment (figure 3*d–h*). Additionally, a strong reduction in FOXO mRNA in wing buds was found in dsFOXO-treated instar 5 nymphs, showing that the injected dsRNA is able to knock down gene expression in the wing bud tissue and that the effect lasts from the time point of injection, in this case from instar 3, until instar 5, which is the stage when adult wing development occurs. As an additional control to validate our RNAi protocol, we injected dsRNA against the developmentally important gene *distalless* (*dll*, electronic supplementary material, figure S6). Wing bud phenotypes in *dll* RNAi-treated individuals were observed before in a different water strider species (*Limnoporus dissortis*, [30]), in which the hind wing bud protrudes from under the abnormally short forewing bud. We found a very similar phenotype in the *dll* RNAi-treated *G. buenoi* individuals (electronic supplementary material, figure S6). In addition to shortened forewing buds, these individuals also had abnormal appendages and experienced problems during moulting. Overall, our results show that, contrary to findings in several other insect species, knockdown of components in the main IIS pathway does not alter wing morph frequencies in *G. buenoi*.

### (e) Forkhead box O RNAi causes an increase in body size but not wing size

As we did not detect differences in wing morph determination in dsGFP-treated individuals and those with an altered IIS pathway, we explored whether manipulated individuals showed differences in body and wing size within wing morph categories. Body size differences are expected in IIS-manipulated animals, given the amount of evidence of size regulation orchestrated by the IIS pathway [31]. We found that dsFOXO individuals in both wing morphs and sexes were significantly larger than dsGFP individuals (4.2% and 4.3% longer for females and males, respectively, electronic supplementary material, figure S7). Thus, FOXO appears to act as a general negative regulator of body size in *G. buenoi*, but interestingly, the increased body size in dsFOXO-treated males and females were not matched by an effect on wing length in either micropterous or macropterous individuals (electronic supplementary material, figure S7).

We also found significant differences in body and wing length between dsINR1-like females and dsGFP females, and between dsINR1 males and dsGFP males, within morph categories (electronic supplementary material, figure S7). However, it should be noted that the reduced body size in dsINR1 and dsINR1-like individuals cannot reliably be attributed to the effect of RNAi against these respective genes, as there is significant cross-knockdown of all three InRs in these treatments (figure 3*d–f*).

## 4. Discussion

Studies of wing polymorphism have a long history and have contributed to our understanding of life-history trade-offs and the adaptive strategies used by organisms in changing

environments [5,8,32–36]. In water striders, there is substantial variation between species in the mechanisms generating wing polymorphism [8,16–19,25]. Our findings establish that photoperiod is the main environmental cue that determines wing morph in *G. buenoi*, as has been suggested before [19]. Furthermore, we showed that a shift from short (12 L : 12 D) to a long (18 L : 6 D) photoperiod is highly inductive of the micropterous morph in a developmental window variably expressed from at least instar 3 until at latest 2 days into instar 5. By contrast, a shift from long (18 L : 6 D) to a short (12 L : 12 D) photoperiod did not seem to have a similar inductive effect (discussed more in detail below). Variation in nutrition had no effect on wing morph induction but increasing density was associated with an increase of macropterous morphs. Taken together, our results are in line with previous evidence [19,24] that wing morph determination in *G. buenoi* is an environmentally induced polyphenic trait and that photoperiod acts as the main environmental switch. Interestingly, unlike in other environmentally induced wing polyphenisms [14,15,37], our RNAi experiments demonstrate that genes in the IIS pathway are not involved in the genetic mechanisms underlying this switch.

## (a) Photoperiod and nymphal rearing density but not nutrition control wing polyphenism in *Gerris buenoi*

By exposing nymphs throughout development to a range of photoperiods, we have demonstrated a strong relationship between (constant) photoperiod and wing morph frequency in *G. buenoi*. Previous work on the role of environmental induction of wing morph in *G. buenoi* was performed in field enclosures and focused on patterns of wing morph frequencies in relation to nymphal stage at the summer solstice [19]. Spence [19] found that no individuals which reached the adult stage before the summer solstice became macropterous, but after the solstice, there was a step-wise increase in the frequency of macropters among individuals in increasingly younger stages at solstice. All individuals that were either instar 1 or 2 at the solstice developed to the macropterous morph [19]. These results suggest that the direction of change in photoperiod is an important cue for wing morph determination in *G. buenoi*, as it appears to be also in the closely related *Gerris odontogaster* [20].

Nonetheless, this interpretation is not entirely satisfactory because the Alberta population of *G. buenoi* produced 100% macropters in the laboratory under 16 L : 8 D constant photoperiod [19], but a mix of macropterous and micropterous morphs at 19 L : 5 D [24]. Although it is tempting to suspect that direction of change in day length is important, changes in photoperiod around the solstice are very small in relation to the developmental duration of late instars of *G. buenoi* in field conditions (roughly 10 days on average for instar 4 and 5 combined, [19]). Under these circumstances, there is little opportunity for a switch to operate on decreasing photoperiod to generate the macropterous individuals seen in fig. 4 in [19]. Thus, the exact contribution of either absolute photoperiod or changing photoperiod on induction during natural conditions remains to be thoroughly understood. Nonetheless, it is clear that at least in laboratory conditions, *G. buenoi* can use absolute photoperiod as a cue for the wing morph switch.

One particularly interesting finding from our work is that females were less likely to become macropterous than males when held at 15 L : 9 D. A similar tendency has been shown before under field conditions [19], where *G. buenoi* females consistently showed lower frequency of macropters than males. We also found a female bias in morph induction in response to high density, where females kept at high density had a higher frequency of macropterous individuals than males (figure 2*a*). Interestingly, Han [25] recently showed that wing morph induction in the gerrid *Tenagogerris euphrosyne* is sex-biased, with females equally likely to become macropters as males at low density but that females were less likely to become macropterous than males at high density. Taken together, these results suggest that the threshold for induction of wing morphs can be sexually dimorphic in water striders and that the direction of the response can differ between species. For a detailed discussion on the adaptive significance of sex-specific wing morph determination see [25].

In line with several other studies on gerromorphans [23,25,38], we found that higher juvenile density led to increased frequency of macropterous individuals (figure 2). This is in contrast with results from Harada and Spence [24] who found that high density increased the frequency of micropterous *G. buenoi* individuals. It is not immediately clear why, although it is possible that differences in photoperiods between our study and that of Harada and Spence [24] played a role. Increased macropter frequencies in response to high nymphal density is likely an adaptive response to enable dispersal when conditions might become detrimental in the future, e.g. due to competition for food. However, the induction of micropters in response to crowding may emphasize that the morph induction can be rather complex; see [24] for a more comprehensive discussion on life-history strategies in response to crowding conditions in water striders.

When we exposed individuals to restricted nutrition regimes of different magnitudes, we found that food availability did not directly influence wing morph determination, at least not in the photoperiods we used (figure 2*b,c*). The low food regime was associated with increased mortality, longer developmental duration and smaller body size. Although no direct effects of nutrition were found on wing morph induction in laboratory conditions, indirect effects may be evident in natural conditions. For example, a longer developmental time may push the development of individuals across solstice and into conditions that favour the development of macropterous morphs.

From the experiments with constant photoperiod, we show that exposure to a long photoperiod of at least 15 h throughout development resulted in some micropterous development, whereas exposure to shorter photoperiods with either 12 or 14 h of daylight resulted strictly in macropterous development. To explore the sensitive stages of wing morph determination in *G. buenoi*, we exposed nymphs to a shift in photoperiod at different developmental stages, similar to an experimental design reported by Inoue & Harada [28] in a study of *A. paludum*. For our population, exposure to a long photoperiod (18 L : 6 D) was highly inductive of the micropterous morph and the sensitive window for this induction lasts from at least instar 3 until latest day 2 in instar 5. Furthermore, our results show that the growth of the adult wings, which occurs in instar 5, can by some mechanism be stunted by photoperiod cues that are received as early as instar 3. It should be noted that these results may

be specific for the tested conditions, especially considering that 12 L : 12 D and 18 L : 6 D and the 6 h shifts in photoperiod is extreme and not experienced by individuals in natural conditions. Nevertheless, these experiments provide valuable information on the potential limits of induction, including the responsiveness of the mechanism that relay information of photoperiod to the growing adult wing tissue. We suggest that the pattern of commitment to wing morph we observe is in line with a view that the default developmental trajectory is towards the macropterous morph, but that this development can be arrested by an unknown mechanism induced by a long photoperiod during a rather long sensitive window, and lead to micropterous development. Interestingly, there is variation in wing morphs within some experimental groups (e.g. I5E and I5M in the 12 L : 12 D to 18 L : 6 D shift, figure 1e), which suggests genetic variation in the sensitive stage for induction.

## (b) Insulin/insulin-like receptor signalling pathway is not involved in wing polyphenism in *Gerris buenoi*

Multiple studies have implicated the IIS pathway as an important driver in wing length polyphenism in several insect species [14,15,37]. Our data, however, suggest that IIS does not play a role in wing polyphenism in *G. buenoi*. This may be because our RNAi knockdown did not reduce protein expression enough to see an effect, or because pathways other than IIS regulate the switch between wing morphs in *G. buenoi*. We argue that the latter alternative is the most-likely explanation based on the following observations. (i) There is no clear link between wing morph determination and nutritional availability in *G. buenoi*. (ii) As expected from the conserved role of IIS in growth regulation, manipulation of IIS components resulted in phenotypic effects similar to the effect of restricted nutrition (e.g. smaller body size), demonstrating that our RNAi treatments induced phenotypic alterations. (iii) The expression of all genes we tested was significantly reduced in individuals that were treated with the corresponding dsRNA, and the knockdown levels were of similar or higher effect sizes compared to similar studies that have reported positive results (e.g. [5,10,12]).

We note, however, that there was significant cross-knockdown among the insulin receptors, so that each InR dsRNA produced a significant knockdown of all InR genes. This is not surprising as the three expressed mRNAs share a considerable amount of sequence identity (approx. 41%, electronic supplementary material, table S5). In principle, cross-knockdown is not desirable, but since we do not see any change in wing morph frequencies in any InR dsRNA treatment, we argue that the observed cross-knockdown should not alter our conclusion.

It is possible that no wing morph switches were induced because the InRs are redundant, meaning that they all can serve as growth regulators of wing development. However, we detected a strong reduction in FOXO mRNA expression in wing buds of dsFOXO-treated individuals. Not only does the FOXO KD show that RNAi works efficiently in wing buds but also that normal levels of FOXO are not required for switching between wing morphs. As FOXO is the main IIS transcriptional output component [31], we interpret the robust reduction of FOXO mRNA in both wing buds and body in our RNAi experiment as our strongest evidence

to suggest that the role of the nutrient-sensing IIS pathway is not involved in *G. buenoi* and thus that the IIS is not ubiquitous in regulating insect wing polyphenisms.

Our results therefore suggest that *G. buenoi* use a different genetic pathway to regulate wing morph differentiation than at least two other hemipterans with wing polyphenisms, the brown planthopper [14] and the soapberry bug [15] in which the IIS pathway controls wing morph determination. We speculate that a potential explanation could be higher environmental stochasticity faced by *G. buenoi* compared to the other species in terms of nutrient availability. The environmental cues for morph determination in the brown planthopper and soapberry bug are associated with the quality of their respective host plants, sugar content [12] and seeds available [15], respectively. The degree of signalling of the nutrient-sensitive IIS pathway thus provides a direct link between environmental cue and growth/induction. *G. buenoi* and other gerrids opportunistically consume a variety of arthropods that fall onto the water surface and thus are subjected to more day to day stochasticity in nutrient availability than species that follow the phenology of a host plant. This view is in line with previous discussions suggesting that nutrition is not a likely factor for wing morph determination in gerrids [33]. The stochastic nature of nutrient availability may thus have favoured other genetic pathways than IIS to be coupled to the wing regulatory network in the evolution of wing polyphenism in *G. buenoi*, and possibly also in other gerrids.

It is not surprising to find that environmental induction can occur with different mechanisms in different species, given that many different genes and pathways can be involved in wing shape and size regulation in insects [39]. In *G. buenoi*, it is clear that the genetic pathway responsible for control of wing length has evolved to be under the influence of photoperiod and will be very interesting to study further.

## 5. Conclusions

Through experimental manipulation of photoperiod, nutrition and density, we have demonstrated that photoperiod is the main environmental cue used to determine different wing morphs in *G. buenoi*. Thus, wing morph variation in this species is an example of an environmentally induced polyphenism. The genetic basis of this polyphenism remains unknown, unlike studies on other environmentally induced wing polyphenisms where the IIS pathway is involved [14,15,37], our RNAi experiments demonstrate that knock down of genes in the IIS pathway do not alter induction of the different wing morphs. Thus, the nutrient-sensing IIS pathway is not ubiquitous in regulating insect wing polyphenism and our results therefore suggest that there are multiple genetic origins to how environmental cues are incorporated and result in wing polyphenism in insects.

Data accessibility. The data used in this manuscript can be found in the file named 'Data_file_Gudmunds_etal_20211221.xslx'. To analyse these data, the R-script 'R_code_Gudmunds_etal_20211221.R' can be used. Available from the Dryad Digital Repository: https://doi.org/10.5061/dryad.zs7h44jbz [40].

The data are provided in the electronic supplementary material [41].

Authors' contributions. E.G.: data curation, formal analysis, investigation, methodology, project administration, validation, visualization, writing—original draft and writing—review and editing; S.N.: formal analysis, investigation, methodology and writing—review and

editing; E.L.: data curation, investigation, methodology and writing—review and editing; M.D.: data curation, investigation, methodology and writing—review and editing; A.K.: methodology, resources, supervision and writing—review and editing; A.H.: conceptualization, formal analysis, funding acquisition, investigation, methodology, project administration, supervision, writing—original draft and writing—review and editing.

All authors gave final approval for publication and agreed to be held accountable for the work performed therein.

Conflict of interest declaration. We declare we have no competing interests.

Funding. This study was supported by funding from the Swedish Research Council to A.H. (grant no. 2020-03349) and Stiftelsen Lars Hiertas Minne (grant no. FO2019-0026) to E.G.

Acknowledgements. We are grateful to Kevin Nielsen who helped with the shifting experiment and to Mattias Vass for help with water strider maintenance. We are grateful to David Armisen and Jesper Boman for comments on the manuscript and Severine Viala for sharing initial density data. We also thank John Spence, Chang Han, the associate editor and one anonymous reviewer for comments that greatly improved the manuscript.

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
