## [Peer Review File · Proceedings of the Royal Society B: Biological Sciences]

Review History

RSPB-2021-1605.R0 (Original submission)

Review form: Reviewer 1 (John Spence)

Recommendation

Major revision is needed (please make suggestions in comments)

Scientific importance: Is the manuscript an original and important contribution to its field?

Marginal

General interest: Is the paper of sufficient general interest?

Acceptable

Quality of the paper: Is the overall quality of the paper suitable?

Poor

Is the length of the paper justified?

Yes

Should the paper be seen by a specialist statistical reviewer?

No

Do you have any concerns about statistical analyses in this paper? If so, please specify them explicitly in your report.

No

It is a condition of publication that authors make their supporting data, code and materials available - either as supplementary material or hosted in an external repository. Please rate, if applicable, the supporting data on the following criteria.

Is it accessible?

Yes

Is it clear?

N/A

Is it adequate?

N/A

Do you have any ethical concerns with this paper?

No

Comments to the Author

I found this work of much interest, and think that it can be turned into a useful, and likely more easily understood, publication about wing-dimorphism in *Gerris buenoi*. Having done some work with this wonderful species, I agree that this *G. buenoi* could be a most appropriate model system for exploring wing polyphenisms in semi-aquatic bugs in a way that can likely improve general understanding of this life-history phenomenon. In the attached Word file edited in 'track changes', I offer suggestions about many specific points that I think merit attention in improving this paper. In addition to amplifying some of these here, I comment about several 'big-picture' issues that seem germane to effective presentation of this work, but which are not much considered in the context of the manuscript. Including them as part of the review, I hope, will prompt the authors to consider them. If they do, IMO, they can help much with developing a more effective and accurate presentation of the story.

1. Interpreting photoperiod effects. I maintain that it is important, in this sort of work, to choose one's photoperiods in clear relation to the source of the population that one is working with. Thus, it is not clear to me why the two photoperiods employed for use in these experiments were chosen, or how this choice helps the argument that photoperiod actually drives induction of wing-morph in nature. I believe that photoperiod is most important, but fail to see how the experiments here provide definitive proof more compelling than other work. Here's the problem. Maximum photoperiod (daylight+civil twilight) as solstice in Toronto, Canada is 16h 39m. Minimum photoperiod that 3rd instars are ever likely to see (based on a highly conservative guess of 15 April) is 14h 12m of daylight, and on 20 Sept, likely as late as one could ever find a 3rd or 4th instar on a pond in southern Ontario daylength is about 13h 30m. Essentially, all of the induction action that is of any significance to these bugs in nature happens over a range of 2.5-3.0 hrs of daylight, rather than as a consequence of the sudden 6 h shifts that they were subjected to in the experiments reported in this paper.

In the experiments reported in this paper, two photoperiods 18:6 and 12:12 both with daylengths, respectively, longer and dramatically shorter than will be ever relevant to natural populations of *G. buenoi* in southern Ontario. Furthermore, the way I read it, stock cultures were held at 20:2 for some reason not explained, and it is not stated how long (or for how many generations) they were held under such conditions. Please consider the selection that may have been imposed, and how it might have affected responses that you seek to illuminate. Also, just to raise one last criticism, which I understand can be levied at most of us who do lab work with this business, is that induction brought about by sudden and large shifts (i.e., >2x the full range of photoperiod difference ever seen in nature) in photoperiod may have little to do with how induction actually happens in nature. Under such conditions, it is possible that many links in the process could go

missing because they just were not initiated.

Therefore, although statements about how photoperiod switches might function in relation to LD and SD regimes are reasonable, these findings must be quite cautiously interpreted with respect to understanding mechanisms that operate in nature. This is the reason why in my own fundamental work on these questions (Spence 1989), I worked with enclosed natural populations and focused mainly on understanding critical stages in relation to natural photoperiods in relation to manipulations that I could accomplish. I submit that this provides a much more reasonable basis for inferences about the clear importance of photoperiod in wing-morph induction than does work involving sudden shifts between two constant photoperiods well outside the range of daylengths that the bugs would ever encounter in nature. The important findings wrt to the photoperiod issue in the 1989 study were: 1) 100% of all bugs that reached the imago stage before the solstice, either male or female, were SW; 2) nymphs that were in instars 3-5 (+1 2nd instar) at the solstice gave rise to an increasing proportion of SW imagoes; 3) females were more likely than males to become SW adults in all categories (very interesting adaptive significance in my mind – in fact, one must wonder why males ever give up flight?); and 4) (fortunately) wrt to *Alberta G. buenoi* anyway, the same basic results are obtained whether the induction happens under daily (increasing or decreasing) shifts in photoperiod or under constant photoperiods in the lab. Thus, at least for *G. buenoi* in mid-northern latitudes, one need not be distracted by Kari Vepsäläinen's interesting finding that direction of photoperiod change seems to matter in some Finnish species.

What seemed most significant to me from the 1989 paper is that there is clearly variation in sensitive stage. I suppose that this variation has genetic basis and is subject to selection. I believe that I have shown that it is, albeit in (unfortunately) unpublished work. We were able to select for ± complete expression of either the LW or SW condition under 19:5 (the longest natural photoperiod in Edmonton) in the laboratory using mass cultures. As I recall we got there in 12-15 generations and ran the cultures for >20 lab generations. Now that I have shed my administrative load, retired, and survived and recovered from two surgeries, I hope to get on with putting this work into print. At any rate, the point I am trying to make here and which could be helpful to for the authors, is that considering the critical stage for induction of both wing-morph and breeding status as quite variable allows us to imagine how this species can adjust its life history in time and space. Given such variation, I am quite suspicious of the conclusion in the paper that results show significant asymmetry in induction depending on whether SW-LW or LW-SW transitions are considered. In fact, as explained below (point #3), I think this is rather wrong-headed. Although I understand that your paper has focused more on the mechanism of wing-length (btw ≠ 'wing size') determination, any discussion or speculation about adaptive significance depends on understanding aspects of induction related to its timing.

2. Linkage between breeding status and voltinism. I was quite surprised to find no mention of connections between wing-length and breeding/diapause status in this paper, as this has been a centerpiece of previous work to set gerrid polyphenisms in the context of adaptive strategies. These features seem to be part of an 'adaptive syndrome' similar, but not quite the same, as C. G. Johnson's classic oogenesis-flight syndrome. Clearly, from previous work, it is clear that SW bugs are virtually all direct breeders in *G. buenoi*. In more than 20 years of serious work with this species, and in handling thousands of overwintered individuals, I have only ever encountered ONE SW individual among overwintered populations! The situation for LW bugs is more interesting because some breed directly while others must traverse and overwintering diapause before breeding in the next year. I am guessing that all bugs that developed under 20:2 in the stock cultures were direct breeders. Were they? Is there any way to know based on records from early cultures? What was the proportion of SW adults in these source cultures? The argument that the mix of morphs resulted from some conscious choice of rearing density seems dubious to me, and I emphasize that the results presented in this paper are at odds with those presented by Harada & Spence (2000). I do not argue that one set is right and the other wrong, but rather that in order to really understand density effects we need to understand why these different results were achieved.

Honestly, I personally like the finding reported in this paper that high density leads to more LW bugs better than the contrary proposition, which is what we found, but I just cannot understand the differences. I just cannot calculate a comparison of densities used because clear information is

not provided for high density treatments in this paper. Presuming that you report starting densities, our low-density treatments were roughly equivalent to yours, but perhaps our high-density treatments were somewhat more dense than yours. However, if high density promotes development of the LW state, one would imagine that we'd have seen different results in Edmonton.

Also, as a tangent here, although it is expressed in the track changes, I must point out that having one bug in a box, no matter what the size of the box, is not a density equivalent to that which includes interactions with other individuals. Social interactions are doubtlessly crucial to the sorts of density effects that one hopes to elucidate with such experiments.

3. The model for adaptive significance. As I digested the paper, it seemed to me that the authors envisaged an altogether different model than I do or that has been employed by others. The discussion and reasoning employed in the paper presupposes that nymphal development is affected by ambient photoperiod from the time of hatching – or perhaps even earlier. With this model in mind, it is logical to expect 'switching' to affect wing development either way. This is not in line with previous thinking, although a rationale for contrast was not presented. Based on the work of Nils Moller Andersen, Kari Vepsäläinen (and his students), me and virtually all others who have worked on this problem, I see it quite differently. I note that all individual *G. buenoi*, whether they go on to eclose as SW or LW development, show development of external wing buds. These, I thought, might be detectable in instar 2 but they are clearly visible as slight swelling in most instar 3 nymphs and, of course, are dead-obvious in all instar 4 and 5 nymphs. Despite a semi-serious effort I could never see differences in wing-pad development, even in instar 5, between bugs that went on to become either SW or LW imagoes. Thus, early lives of all developing nymphs include some developmental progress toward developing wings.

IMO, there is, for each individual (with variation in the population), one critical period during development, during which the investment in full-wings and at least some of the associated machinery can be turned off. If it is not turned off at that time, normal wing and muscle development continues. Nymphs destined to be SW females (and perhaps even males) have the possibility of shifting investment into development of gonads and (in females) to oogenesis. By doing so, they can begin oviposition as soon as they can mate, shortening the pre-reproductive period. Males can put all of their energy into securing mates and inseminating as many females as is possible. For individuals that eclose as LW adults there is a most interesting separation of effect between regulation of genes that control wing-morph and breeding status.

This separation, IMO, is most deserving of study. For SW bugs these decisions are linked and seem to be (or may be) controlled by the same photoperiod switch. For *G. buenoi* in Alberta, however, an additional decision is required for nymphs that progress to LW adults. Direct breeders, develop flight muscles, which can then be histolysed to meet energy needs for reproduction after flight (this incidentally happens also in overwintered bugs that have traversed diapause). Diapause-bound bugs develop full flight capability but simply will not breed until after diapause. One can tell diapause-bound bugs from direct breeders by looking at pigmentation of the ventral abdominal segments – diapause bugs will be dark, direct breeders pale. Another sort of switch happens here because the earliest emerging LW bugs will be direct breeders, but these cohorts will be followed in short order by diapause-bound cohorts until the ices comes – although generally, all *G. buenoi* are gone from the ponds by mid-Sept in Alberta. I will emphasize one last point here that relates to an error in your paper. You describe *G. buenoi* as being bi-voltine in Canada. This may be true for a few of the southern-most populations, but even there I am doubtful. It has certainly not been studied. Here in Alberta, I can say with certainty that *G. buenoi* is partially bi-voltine, i.e., overwintered females split their reproductive effort between direct breeding and diapause-bound progeny. This interesting and critical aspect of life-history is regulated by the processes discussed above.

Given my understanding, which I think is very well supported by previous work in both *G. buenoi* and other gerrid species, I think it is simply 'wrong-headed' to think of bugs initially reared under short-day photoperiods as being programmed to be LW and those reared in early stages under long-day photoperiod to be programmed to be SW. I submit that there is one two-way switch that operates on bugs that are all developing toward being LW adults, and the setting of this switch may be changed by photoperiod experienced at a particular critical stage, the timing of which varies among individuals. There are no nymphs set to develop as SW adults in

the early stages of post-embryonic development. IMO, this is not a semantic argument. The paper, as written, rather directly assumes a different chain of events. I believe that this contradicts the best available information, and that if the authors, want to use that assumption anyway in the face of long-established thinking, that they must directly say why and show that the assumption is valid. It took me awhile to understand what was bothering me about the analysis that was presented in the paper, but I have finally understood that this was a most bothersome aspect.

4. Scientific format. I try not to be obsessed with matters of form in reading scientific papers, but in the present manuscript, I find that too much of what should have been in the Methods section ends up in the first part of each of the sections meant to present the Results. Also, I note that the results include some matters that are properly presented in the Discussion. I used to ask my own student to keep a check on this by looking very carefully at matters requiring citation in the Results section. In general, they should not be there. What is most critically missing in the manuscript is a clear explanation of the methods used that is well integrated with a clear statement of the hypotheses being addressed specifically by each thrust of the research. Also, I personally, struggled with the use of molecular biology methods here that were not explained, but simply described as jargon. If the authors want readers who are interested this work to generally understand what was done, I think that it's rationale and execution must be more clearly explained. I was left in the position of having to simply accept the interesting conclusions as assertions without really understanding their basis. I've tried to help with specific suggestions about how things could be improved in the MS edited in tracked-changes. I think this should not pose much of a challenge to the authors, but that such revision will considerably improve how much the work can be appreciated by readers.

5. Literature that might help. I found that 3 papers quite germane, and I think critical to proper understanding of this work, were missing from the Literature Cited. These are as follows. Two are from my own work dealing directly with *G. buenoi*, and one is a critical foundational paper by Nils Møller Andersen. Although the latter paper is about a couple of Danish species it was among the first to lay a clear foundation for appreciating the existence of wing-polyphenism in gerrids. If you have difficulty in finding these papers, I will happily send pdfs of them in response to request to jrspen@gmail.com.

Andersen N.M. 1973: Seasonal polymorphism and developmental changes in organs of flight and reproduction in bivoltine pondskaters (Hem.: Gerridae). *Entomol. Scand.* 4: 1-20.

Spence, JR. 2000. Seasonal aspects of flight in water striders (Hemiptera: Gerridae). *Entomological Science* 3: 399-417.

Harada, T & JR Spence. 2000. Nymphal density and life histories of two water striders (Hemiptera: Gerridae). *The Canadian Entomologist* 132: 353-364.

I think that 5 general issues raised above cover the high points of my general overall reaction to the manuscript. Please see the detailed editing for more specific suggestions for revision. I emphasize in closing that I am willing to assist the authors further should they desire additional input.

John Spence, Edmonton, Alberta, CANADA

Review form: Reviewer 2 (Chang Han)

Recommendation

Major revision is needed (please make suggestions in comments)

Scientific importance: Is the manuscript an original and important contribution to its field?

Good

General interest: Is the paper of sufficient general interest?

Good

Quality of the paper: Is the overall quality of the paper suitable?

Good

Is the length of the paper justified?

Yes

Should the paper be seen by a specialist statistical reviewer?

No

Do you have any concerns about statistical analyses in this paper? If so, please specify them explicitly in your report.

No

It is a condition of publication that authors make their supporting data, code and materials available - either as supplementary material or hosted in an external repository. Please rate, if applicable, the supporting data on the following criteria.

Is it accessible?

Yes

Is it clear?

Yes

Is it adequate?

Yes

Do you have any ethical concerns with this paper?

No

Comments to the Author

This study is showing that photoperiod is the most important environmental factor to shape wing polymorphism in a water strider *Gerris buenoi*, but the wing expression is independent from nutrient sensing pathways. The manuscript yielded some interesting results and would have required considerable work. Yet, I have some comments that should be addressed. I apologise if any criticisms stem from misunderstanding of what was done and hope at least some comments will be of use to the authors.

Major comments

L94-98 and others. I had the feeling that authors need to introduce a considerable existing literature on the relationship between environmental factors (density, sex ratio, photoperiod, temperature, sex ratio etc) and wing development in water striders. In particular, Tetsuo Harada did lots of works on this issue. In addition, since there are quite a few studies on the genetic basis of wing development in water striders, authors should have discussed them in their manuscript.

L220-221, L228-232, L327-330, L339-341. I think that different trajectories (Figure 2) and density effects (Figure 3a) might be because 18:6 is not a "very" strong photoperiod to induce the expression of single wing morph compared to 12:12. In Figure 1b, 12:12 is a very strong photoperiod to induce the expression of single wing morph (LW) because only LW individuals also developed under 14:10. 18:6 might also be a strong photoperiod to induce the expression of single wing morph (SW). However, because a few LW individuals developed under 16:8 condition, 18:6 might not be a "very" strong photoperiod to induce the expression of single wing morph. If authors estimated switcher probabilities across stages under 20:4, they might have got a pattern similar to the one shown in Figure 2c. I think authors need to discuss this possibility. (However, in my personal opinion, I am also in line with authors' argument that photoperiod is stronger factor to shape gerrid wing polymorphism.)

Also, if authors measured macroptery frequencies across different density conditions under 20:4, they might have found no LW individuals. But, they got LW individuals from the stock population reared under 20:4 and high density, supporting that photoperiod is stronger factor to shape gerrid wing polymorphism.

Minor comments

L20-21. Please add why nutrient sensing pathway is important in the development of wing polymorphism.

L94-98. "Some species" show up a couple of times in this paragraph.

L117-122. Please be more specific about how stock populations are maintained. Experiments were done in Sweden using gerrid populations from Canada. How did they maintain genetic diversity of the stock population?

L119. I missed any explanation of why authors maintained their stock population under such extreme photoperiod condition (20:4) ?

L129-130. I failed to find how authors set starting photoperiods. 18:6 and 12:12? How were photoperiods switched at some developmental stage? Did you increase/decrease photoperiod gradually? or abruptly?

L136, 140, 229 and others. I found that there were lots of places citing wrong figures. Please correct them.

L145-6. How many individuals were kept in high density treatment?

L202-3. I think the appropriate way to show this is to make figure 1b for males and females, respectively. If you revise Figure 1b to show sex specific patterns, I think figure 1c is not necessary.

L212-3. Photoperiod increase was from 12:12 to 18:6, and photoperiod decrease was from 18:6 to 12:12?

L229-230. I missed the test of interactive effects of density and photoperiod on the expression of wing polymorphism. Why did the authors test density effects only at 18:6 photoperiod condition?

L234. Did the food conditions (3,5,7 crickets) affect the developmental duration of nymphs?

Because morphological differences between food treatments in Figure 3c were significant but not so dramatic, food deficiency might not be so stressful for gerrids to develop LW.

L243. I suggest defining what the low and high diet treatments are.

L235-7. Why did you test diet effects only at the extreme photoperiod conditions? I missed diet effects under other photoperiod conditions. Although authors suggested that photoperiod effects were greater than diet effects, the results might be because diet treatments were not so stressful, compared to the extreme photoperiod conditions.

L335. What do phases mean here?

Figure 1b. Please add parental wing morphs for 12:12, 14:10, 16:8 and 18:6 treatments.

Figure 2a. I understood what the abbreviations mean on the x-axis when I read methods. Please add what the abbreviations mean to the legend.

Figure 3b. Please add what 3,5 and 7 mean.

Signed by Chang Han

Review form: Reviewer 3 (Kenneth McKenna)

Recommendation

Accept with minor revision (please list in comments)

Scientific importance: Is the manuscript an original and important contribution to its field?

Excellent

General interest: Is the paper of sufficient general interest?

Excellent

Quality of the paper: Is the overall quality of the paper suitable?

Excellent

Is the length of the paper justified?

Yes

Should the paper be seen by a specialist statistical reviewer?

No

Do you have any concerns about statistical analyses in this paper? If so, please specify them explicitly in your report.

No

It is a condition of publication that authors make their supporting data, code and materials available - either as supplementary material or hosted in an external repository. Please rate, if applicable, the supporting data on the following criteria.

Is it accessible?

Yes

Is it clear?

Yes

Is it adequate?

Yes

Do you have any ethical concerns with this paper?

No

Comments to the Author

The manuscript by Gudmunds et al., attempts to identify what environmental factors determine wing polyphenism in the water strider, *Gerris buenoi*. The manuscript details a robust experimental analysis finding that photoperiod during the 3rd and 4th nymphal instars determines this polyphenism. Additionally, Gudmunds et al., attempt to identify whether photoperiod specifies the short wing morph via insulin signaling, a logical candidate considering most hemimetabolous wing polyphenisms are nutrition-dependent. The authors find however that perturbation of insulin signaling components via RNAi does not induce the short-wing morph therefore suggesting photoperiod may act through a different intrinsic growth regulatory pathway to stunt wing growth in short wing individuals.

Altogether, I think this is a fantastic manuscript and I only have minor requests of the authors.

1. In figure 2 legend, can you write out what I3, I4E, I4L...etc, mean to make it easier for the reader?
2. It would be nice to see an ontogenetic series of the wings from the 3rd nymphal instar to adulthood in 12:12, 18:6, and perhaps the photoperiod switching experiments. Specifically, it would be nice if you can take the wings off the animal, flatten under a microscope slide, and image to scale. I ask this because it seems that instars 3 and 4 are key stages where the growth might stunt, and looking that *Micropterus* wings, it is almost as if growth is stunted but differentiation of the anterior-posterior and proximo-distal axis is not. It would be nice to have a visual representation of this for the readers.
3. This is just a thoughtful note for the authors. I am not requesting any of the following be included in the manuscript. It seems that the short-wing morphs are left with a fully organized, yet vestigial wing. It reminds me of the difference in the forewing and hindwing in Diptera, wherein the hindwing has been reduced to become the haltere. This essentially happens by dampening the growth regulatory effects of virtually all of the morphogens (Pavlopoulos & Akam, 2011; Weatherbee et al., 1998). In normal wing development, the response to morphogens

seems to be linked to the temporal transcriptional turnover directed by rising ecdysone levels (Mirth et al., 2009; Oliveira et al., 2014), and photoperiod has been associated with temporal alterations in ecdysone biosynthesis in many polyphenisms (Nijhout, 1999). I wonder whether this wing polyphenism is determined by affecting ecdysone biosynthesis, and the wings have evolved multiple thresholds for ecdysone, wherein crossing the minimum threshold facilitates the progression of AP and PD patterning, and crossing the next threshold facilitates proliferation and this might be realizable by toying around with the crosstalk between ecdysone signaling and Hippo/Warts signaling (McKenna et al., 2019). This last reference discusses the literature of ecdysone-Hippo/Warts cross talk.

McKenna, K. Z., Tao, D., & Nijhout, H. F. (2019). Exploring the Role of Insulin Signaling in Relative Growth: A Case Study on Wing-Body Scaling in Lepidoptera. *Integrative and Comparative Biology*, 59(5), 1324–1337. <https://doi.org/10.1093/icb/icz080>

Mirth, C. K., Truman, J. W., & Riddiford, L. M. (2009). The Ecdysone receptor controls the post-critical weight switch to nutrition-independent differentiation in *Drosophila* wing imaginal discs. *Development*, 136(14), 2345–2353. <https://doi.org/10.1242/dev.032672>

Nijhout, H. F. (1999). Control Mechanisms of Polyphenic Development in Insects: In polyphenic development, environmental factors alter some aspects of development in an orderly and predictable way. *BioScience*, 49(3), 181–192. <https://doi.org/10.2307/1313508>

Oliveira, M. M., Shingleton, A. W., & Mirth, C. K. (2014). Coordination of Wing and Whole-Body Development at Developmental Milestones Ensures Robustness against Environmental and Physiological Perturbations. *PLOS Genetics*, 10(6), e1004408.

<https://doi.org/10.1371/journal.pgen.1004408>

Pavlopoulos, A., & Akam, M. (2011). Hox gene *Ultrabithorax* regulates distinct sets of target genes at successive stages of *Drosophila* haltere morphogenesis. *Proceedings of the National Academy of Sciences*, 108(7), 2855–2860. <https://doi.org/10.1073/pnas.1015077108>

Weatherbee, S. D., Halder, G., Kim, J., Hudson, A., & Carroll, S. (1998). *Ultrabithorax* regulates genes at several levels of the wing-patterning hierarchy to shape the development of the *Drosophila* haltere. *Genes & Development*, 12(10), 1474–1482.

<https://doi.org/10.1101/gad.12.10.1474>

Decision letter (RSPB-2021-1605.R0)

27-Aug-2021

Dear Mr Gudmunds:

I am writing to inform you that your manuscript RSPB-2021-1605 entitled "Photoperiod controls wing polyphenism in a water strider independently of insulin receptor signaling components" has, in its current form, been rejected for publication in *Proceedings B*.

This action has been taken on the advice of referees, who have recommended that substantial revisions are necessary. With this in mind we would be happy to consider a resubmission, provided the comments of the referees are fully addressed. However please note that this is not a provisional acceptance.

Sincerely,
Dr Sasha Dall
<mailto:proceedingsb@royalsociety.org>

Associate Editor
Board Member: 1
Comments to Author:

The manuscript under consideration was reviewed by three experts and myself. The study tackles the question of environmental factors mediating wing polyphenism in a water strider, through laboratory experiments manipulating temperature, density and nutrition, and using RNA interference to test the role of insulin signalling. All reviewers found the work interesting, as did I.

All reviewers raised important points that I believe should be addressed.

Reviewer 1 suggests an alternative model for wing development: that all individuals begin on a long-winged trajectory and some switch to a short-winged trajectory depending on environmental conditions. This model should be explicitly considered, along with the strength of evidence for whether it's the length of photoperiod in the critical developmental window that matters, or whether photoperiod is increasing or decreasing. Reviewer 1 also suggests stronger links between the manuscript's hypotheses and the literature, including linking hypotheses about wing development to reproductive strategy. Both reviewers 1 and 2 suggest relevant literature that should be cited.

The reviewers have raised important points about the methods that need attention and discussion. Reviewer 1 points out that the photoperiod range tested is greater than natural variation and that a sudden switch is artificial, and Reviewer 2 points out that the short photoperiod appears to be a more extreme treatment than the long photoperiod. Reviewer 1 also notes concerns about the density treatment, in that the low density treatment is an isolation treatment; the reader also need to see the results from low, medium and high treatments separately. It would be helpful if the authors can provide evidence that the low nutritional treatment was stressful enough to provoke a response to the treatment, as Reviewer 2 suggests. Reviewers 1 and 2 ask for clarification about important aspects of the methods, especially the density treatment and the feeding schedule and severity of the low-food treatment, which affects interpretation of the nutrition treatment.

Reviewer(s)' Comments to Author:
Referee: 1

Comments to the Author(s)

I found this work of much interest, and think that it can be turned into a useful, and likely more easily understood, publication about wing-dimorphism in *Gerris buenoi*. Having done some

work with this wonderful species, I agree that this *G. buenoi* could be a most appropriate model system for exploring wing polyphenisms in semi-aquatic bugs in a way that can likely improve general understanding of this life-history phenomenon. In the attached Word file edited in 'track changes', I offer suggestions about many specific points that I think merit attention in improving this paper. In addition to amplifying some of these here, I comment about several 'big-picture' issues that seem germane to effective presentation of this work, but which are not much considered in the context of the manuscript. Including them as part of the review, I hope, will prompt the authors to consider them. If they do, IMO, they can help much with developing a more effective and accurate presentation of the story.

1. Interpreting photoperiod effects. I maintain that it is important, in this sort of work, to choose one's photoperiods in clear relation to the source of the population that one is working with. Thus, it is not clear to me why the two photoperiods employed for use in these experiments were chosen, or how this choice helps the argument that photoperiod actually drives induction of wing-morph in nature. I believe that photoperiod is most important, but fail to see how the experiments here provide definitive proof more compelling than other work. Here's the problem. Maximum photoperiod (daylight+civil twilight) as solstice in Toronto, Canada is 16h 39m. Minimum photoperiod that 3rd instars are ever likely to see (based on a highly conservative guess of 15 April) is 14h 12m of daylight, and on 20 Sept, likely as late as one could ever find a 3rd or 4th instar on a pond in southern Ontario daylength is about 13h 30m. Essentially, all of the induction action that is of any significance to these bugs in nature happens over a range of 2.5-3.0 hrs of daylight, rather than as a consequence of the sudden 6 h shifts that they were subjected to in the experiments reported in this paper.

In the experiments reported in this paper, two photoperiods 18:6 and 12:12 both with daylengths, respectively, longer and dramatically shorter than will be ever relevant to natural populations of *G. buenoi* in southern Ontario. Furthermore, the way I read it, stock cultures were held at 20:2 for some reason not explained, and it is not stated how long (or for how many generations) they were held under such conditions. Please consider the selection that may have been imposed, and how it might have affected responses that you seek to illuminate. Also, just to raise one last criticism, which I understand can be levied at most of us who do lab work with this business, is that induction brought about by sudden and large shifts (i.e., >2x the full range of photoperiod difference ever seen in nature) in photoperiod may have little to do with how induction actually happens in nature. Under such conditions, it is possible that many links in the process could go missing because they just were not initiated.

Therefore, although statements about how photoperiod switches might function in relation to LD and SD regimes are reasonable, these findings must be quite cautiously interpreted with respect to understanding mechanisms that operate in nature. This is the reason why in my own fundamental work on these questions (Spence 1989), I worked with enclosed natural populations and focused mainly on understanding critical stages in relation to natural photoperiods in relation to manipulations that I could accomplish. I submit that this provides a much more reasonable basis for inferences about the clear importance of photoperiod in wing-morph induction than does work involving sudden shifts between two constant photoperiods well outside the range of daylengths that the bugs would ever encounter in nature. The important findings wrt to the photoperiod issue in the 1989 study were: 1) 100% of all bugs that reached the imago stage before the solstice, either male or female, were SW; 2) nymphs that were in instars 3-5 (+1 2nd instar) at the solstice gave rise to an increasing proportion of SW imagoes; 3) females were more likely than males to become SW adults in all categories (very interesting adaptive significance in my mind – in fact, one must wonder why males ever give up flight?); and 4) (fortunately) wrt to Alberta *G. buenoi* anyway, the same basic results are obtained whether the induction happens under daily (increasing or decreasing) shifts in photoperiod or under constant photoperiods in the lab. Thus, at least for *G. buenoi* in mid-northern latitudes, one need not be distracted by Kari Vepsäläinen's interesting finding that direction of photoperiod change seems to matter in some Finnish species.

What seemed most significant to me from the 1989 paper is that there is clearly variation in sensitive stage. I suppose that this variation has genetic basis and is subject to selection. I believe that I have shown that it is, albeit in (unfortunately) unpublished work. We were able to select for ± complete expression of either the LW or SW condition under 19:5 (the longest natural

photoperiod in Edmonton) in the laboratory using mass cultures. As I recall we got there in 12-15 generations and ran the cultures for >20 lab generations. Now that I have shed my administrative load, retired, and survived and recovered from two surgeries, I hope to get on with putting this work into print. At any rate, the point I am trying to make here and which could be helpful to for the authors, is that considering the critical stage for induction of both wing-morph and breeding status as quite variable allows us to imagine how this species can adjust its life history in time and space. Given such variation, I am quite suspicious of the conclusion in the paper that results show significant asymmetry in induction depending on whether SW-LW or LW-SW transitions are considered. In fact, as explained below (point #3), I think this is rather wonng-headed. Although I understand that your paper has focused more on the mechanism of wing-length (btw ≠ 'wing size') determination, any discussion or speculation about adaptive significance depends on understanding aspects of induction related to its timing.

2. Linkage between breeding status and voltinism. I was quite surprised to find no mention of connections between wing-length and breeding/diapause status in this paper, as this has been a centerpiece of previous work to set gerrid polyphenisms in the context of adaptive strategies. These features seem to be part of an 'adaptive syndrome' similar, but not quite the same, as C. G. Johnson's classic oogenesis-flight syndrome. Clearly, from previous work, it is clear that SW bugs are virtually all direct breeders in *G. buenoi*. In more than 20 years of serious work with this species, and in handling thousands of overwintered individuals, I have only ever encountered ONE SW individual among overwintered populations! The situation for LW bugs is more interesting because some breed directly while others must traverse and overwintering diapause before breeding in the next year. I am guessing that all bugs that developed under 20:2 in the stock cultures were direct breeders. Were they? Is there any way to know based on records from early cultures? What was the proportion of SW adults in these source cultures? The argument that the mix of morphs resulted from some conscious choice of rearing density seems dubious to me, and I emphasize that the results presented in this paper are at odds with those presented by Harada & Spence (2000). I do not argue that one set is right and the other wrong, but rather that in order to really understand density effects we need to understand why these different results were achieved.

Honestly, I personally like the finding reported in this paper that high density leads to more LW bugs better than the contrary proposition, which is what we found, but I just cannot understand the differences. I just cannot calculate a comparison of densities used because clear information is not provided for high density treatments in this paper. Presuming that you report starting densities, our low-density treatments were roughly equivalent to yours, but perhaps our high-density treatments were somewhat more dense than yours. However, if high density promotes development of the LW state, one would imagine that we'd have seen different results in Edmonton.

Also, as a tangent here, although it is expressed in the track changes, I must point out that having one bug in a box, no matter what the size of the box, is not a density equivalent to that which includes interactions with other individuals. Social interactions are doubtlessly crucial to the sorts of density effects that one hopes to elucidate with such experiments.

3. The model for adaptive significance. As I digested the paper, it seemed to me that the authors envisaged an altogether different model than I do or that has been employed by others. The discussion and reasoning employed in the paper presupposes that nymphal development is affected by ambient photoperiod from the time or hatching – or perhaps even earlier. With this model in mind, it is logical to expect 'switching' to affect wing development either way. This is not in line with previous thinking, although a rationale for contrast was not presented. Based on the work of Nils Moller Andersen, Kari Vepsäläinen (and is students), me and virtually all others who have worked on this problem, I see it quite differently. I note that all individual *G. buenoi*, whether they go onto eclose as SW or LW development, show development of external wing buds. These, I thought, might be detectable in instar 2 but they are clearly visible as slight swelling in most instar 3 nymphs and, of course, are dead-obvious in all instar 4 and 5 nymphs. Despite a semi-serious effort I could never see differences in wing-pad development, even in instar 5, between bugs that went on to become either SW or LW imagoes. Thus, early lives of all developing nymphs include some developmental progress toward developing wings.

IMO, there is, for each individual (with variation in the population), one critical period during development, during which the investment in full-wings and at least some of the associated machinery can be turned off. If it is not turned off at that time, normal wing and muscle development continues. Nymphs destined to be SW females (and perhaps even males) have the possibility of shifting investment into development of gonads and (in females) to oogenesis. By doing so, they can begin oviposition as soon as they can mate, shortening the pre-reproductive period. Males can put all of their energy into securing mates and inseminating as many females as is possible. For individuals that eclose as LW adults there is a most interesting separation of effect between regulation of genes that control wing-morph and breeding status.

This separation, IMO, is most deserving of study. For SW bugs these decisions are linked and seem to be (or may be) controlled by the same photoperiod switch. For *G. buenoi* in Alberta, however, an additional decision is required for nymphs that progress to LW adults. Direct breeders, develop flight muscles, which can then be histolysed to meet energy needs for reproduction after flight (this incidentally happens also in overwintered bugs that have traversed diapause). Diapause-bound bugs develop full flight capability but simply will not breed until after diapause. One can tell diapause-bound bugs from direct breeders by looking at pigmentation of the ventral abdominal segments – diapause bugs will be dark, direct breeders pale. Another sort of switch happens here because the earliest emerging LW bugs will be direct breeders, but these cohorts will be followed in short order by diapause-bound cohorts until the ices comes – although generally, all *G. buenoi* are gone from the ponds by mid-Sept in Alberta. I will emphasize one last point here that relates to an error in your paper. You describe *G. buenoi* as being bi-voltine in Canada. This may be true for a few of the southern-most populations, but even there I am doubtful. It has certainly not been studied. Here in Alberta, I can say with certainty that *G. buenoi* is partially bi-voltine, i.e., overwintered females split their reproductive effort between direct breeding and diapause-bound progeny. This interesting and critical aspect of life-history is regulated by the processes discussed above.

Given my understanding, which I think is very well supported by previous work in both *G. buenoi* and other gerrid species, I think it is simply ‘wrong-headed’ to think of bugs initially reared under short-day photoperiods as being programmed to be LW and those reared in early stages under long-day photoperiod to be programmed to be SW. I submit that there is one two-way switch that operates on bugs that are all developing toward being LW adults, and the setting of this switch may be changed by photoperiod experienced at a particular critical stage, the timing of which varies among individuals. There are no nymphs set to develop as SW adults in the early stages of post-embryonic development. IMO, this is not a semantic argument. The paper, as written, rather directly assumes a different chain of events. I believe that this contradicts the best available information, and that if the authors, want to use that assumption anyway in the face of long-established thinking, that they must directly say why and show that the assumption is valid. It took me awhile to understand what was bothering me about the analysis that was presented in the paper, but I have finally understood that this was a most bothersome aspect.

4. Scientific format. I try not to be obsessed with matters of form in reading scientific papers, but in the present manuscript, I find that too much of what should have been in the Methods section ends up in the first part of each of the sections meant to present the Results. Also, I note that the results include some matters that are properly presented in the Discussion. I used to ask my own student to keep a check on this by looking very carefully at matters requiring citation in the Results section. In general, they should not be there. What is most critically missing in the manuscript is a clear explanation of the methods used that is well integrated with a clear statement of the hypotheses being addressed specifically by each thrust of the research. Also, I personally, struggled with the use of molecular biology methods here that were not explained, but simply described as jargon. If the authors want readers who are interested this work to generally understand what was done, I think that it's rationale and execution must be more clearly explained. I was left in the position of having to simply accept the interesting conclusions as assertions without really understanding their basis. I've tried to help with specific suggestions about how things could be improved in the MS edited in tracked-changes. I think this should not pose much of a challenge to the authors, but that such revision will considerably improve how much the work can be appreciated by readers.

5. Literature that might help. I found that 3 papers quite germane, and I think critical to proper understanding of this work, were missing from the Literature Cited. These are as follows. Two are from my own work dealing directly with *G. buenoi*, and one is a critical foundational paper by Nils Møller Andersen. Although the latter paper is about a couple of Danish species it was among the first to lay a clear foundation for appreciating the existence of wing-polyphenism in gerrids. If you have difficulty in finding these papers, I will happily send pdfs of them in response to request to jrspen@gmail.com.

Andersen N.M. 1973: Seasonal polymorphism and developmental changes in organs of flight and reproduction in bivoltine pondskaters (Hem.: Gerridae). *Entomol. Scand.* 4: 1-20.

Spence, JR. 2000. Seasonal aspects of flight in water striders (Hemiptera: Gerridae). *Entomological Science* 3: 399-417.

Harada, T & JR Spence. 2000. Nymphal density and life histories of two water striders (Hemiptera: Gerridae). *The Canadian Entomologist* 132: 353-364.

I think that 5 general issues raised above cover the high points of my general overall reaction to the manuscript. Please see the detailed editing for more specific suggestions for revision. I emphasize in closing that I am willing to assist the authors further should they desire additional input.

John Spence, Edmonton, Alberta, CANADA

Referee: 2

Comments to the Author(s)

This study is showing that photoperiod is the most important environmental factor to shape wing polymorphism in a water strider *Gerris buenoi*, but the wing expression is independent from nutrient sensing pathways. The manuscript yielded some interesting results and would have required considerable work. Yet, I have some comments that should be addressed. I apologise if any criticisms stem from misunderstanding of what was done and hope at least some comments will be of use to the authors.

Major comments

L94-98 and others. I had the feeling that authors need to introduce a considerable existing literature on the relationship between environmental factors (density, sex ratio, photoperiod, temperature, sex ratio etc) and wing development in water striders. In particular, Tetsuo Harada did lots of works on this issue. In addition, since there are quite a few studies on the genetic basis of wing development in water striders, authors should have discussed them in their manuscript.

L220-221, L228-232, L327-330, L339-341. I think that different trajectories (Figure 2) and density effects (Figure 3a) might be because 18:6 is not a "very" strong photoperiod to induce the expression of single wing morph compared to 12:12. In Figure 1b, 12:12 is a very strong photoperiod to induce the expression of single wing morph (LW) because only LW individuals also developed under 14:10. 18:6 might also be a strong photoperiod to induce the expression of single wing morph (SW). However, because a few LW individuals developed under 16:8 condition, 18:6 might not be a "very" strong photoperiod to induce the expression of single wing morph. If authors estimated switcher probabilities across stages under 20:4, they might have got a pattern similar to the one shown in Figure 2c. I think authors need to discuss this possibility. (However, in my personal opinion, I am also in line with authors' argument that photoperiod is stronger factor to shape gerrid wing polymorphism.)

Also, if authors measured macroptery frequencies across different density conditions under 20:4, they might have found no LW individuals. But, they got LW individuals from the stock population reared under 20:4 and high density, supporting that photoperiod is stronger factor to shape gerrid wing polymorphism.

Minor comments

L20-21. Please add why nutrient sensing pathway is important in the development of wing polymorphism.

L94-98. "Some species" show up a couple of times in this paragraph.

L117-122. Please be more specific about how stock populations are maintained. Experiments were done in Sweden using gerrid populations from Canada. How did they maintain genetic diversity of the stock population?

L119. I missed any explanation of why authors maintained their stock population under such extreme photoperiod condition (20:4) ?

L129-130. I failed to find how authors set starting photoperiods. 18:6 and 12:12? How were photoperiods switched at some developmental stage? Did you increase/decrease photoperiod gradually? or abruptly?

L136, 140, 229 and others. I found that there were lots of places citing wrong figures. Please correct them.

L145-6. How many individuals were kept in high density treatment?

L202-3. I think the appropriate way to show this is to make figure 1b for males and females, respectively. If you revise Figure 1b to show sex specific patterns, I think figure 1c is not necessary.

L212-3. Photoperiod increase was from 12:12 to 18:6, and photoperiod decrease was from 18:6 to 12:12?

L229-230. I missed the test of interactive effects of density and photoperiod on the expression of wing polymorphism. Why did the authors test density effects only at 18:6 photoperiod condition?

L234. Did the food conditions (3,5,7 crickets) affect the developmental duration of nymphs?

Because morphological differences between food treatments in Figure 3c were significant but not so dramatic, food deficiency might not be so stressful for gerrids to develop LW.

L243. I suggest defining what the low and high diet treatments are.

L235-7. Why did you test diet effects only at the extreme photoperiod conditions? I missed diet effects under other photoperiod conditions. Although authors suggested that photoperiod effects were greater than diet effects, the results might be because diet treatments were not so stressful, compared to the extreme photoperiod conditions.

L335. What do phases mean here?

Figure 1b. Please add parental wing morphs for 12:12, 14:10, 16:8 and 18:6 treatments.

Figure 2a. I understood what the abbreviations mean on the x-axis when I read methods. Please add what the abbreviations mean to the legend.

Figure 3b. Please add what 3,5 and 7 mean.

Signed by Chang Han

Referee: 3

Comments to the Author(s)

The manuscript by Gudmunds et al., attempts to identify what environmental factors determine wing polyphenism in the water strider, *Gerris buenoi*. The manuscript details a robust experimental analysis finding that photoperiod during the 3rd and 4th nymphal instars determines this polyphenism. Additionally, Gudmunds et al., attempt to identify whether photoperiod specifies the short wing morph via insulin signaling, a logical candidate considering most hemimetabolous wing polyphenisms are nutrition-dependent. The authors find however that perturbation of insulin signaling components via RNAi does not induce the short-wing morph therefore suggesting photoperiod may act through a different intrinsic growth regulatory pathway to stunt wing growth in short wing individuals.

Altogether, I think this is a fantastic manuscript and I only have minor requests of the authors.

1. In figure 2 legend, can you write out what I3, I4E, I4L...etc, mean to make it easier for the reader?

2. It would be nice to see an ontogenetic series of the wings from the 3rd nymphal instar to adulthood in 12:12, 18:6, and perhaps the photoperiod switching experiments. Specifically, it would be nice if you can take the wings off the animal, flatten under a microscope slide, and

image to scale. I ask this because it seems that instars 3 and 4 are key stages where the growth might stunt, and looking that *Micropterus* wings, it is almost as if growth is stunted but differentiation of the anterior-posterior and proximo-distal axis is not. It would be nice to have a visual representation of this for the readers.

3. This is just a thoughtful note for the authors. I am not requesting any of the following be included in the manuscript. It seems that the short-wing morphs are left with a fully organized, yet vestigial wing. It reminds me of the difference in the forewing and hindwing in Diptera, wherein the hindwing has been reduced to become the haltere. This essentially happens by dampening the growth regulatory effects of virtually all of the morphogens (Pavlopoulos & Akam, 2011; Weatherbee et al., 1998). In normal wing development, the response to morphogens seems to be linked to the temporal transcriptional turnover directed by rising ecdysone levels (Mirth et al., 2009; Oliveira et al., 2014), and photoperiod has been associated with temporal alterations in ecdysone biosynthesis in many polyphenisms (Nijhout, 1999). I wonder whether this wing polyphenism is determined by affecting ecdysone biosynthesis, and the wings have evolved multiple thresholds for ecdysone, wherein crossing the minimum threshold facilitates the progression of AP and PD patterning, and crossing the next threshold facilitates proliferation and this might be realizable by toying around with the crosstalk between ecdysone signaling and Hippo/Warts signaling (McKenna et al., 2019). This last reference discusses the literature of ecdysone-Hippo/Warts cross talk.

McKenna, K. Z., Tao, D., & Nijhout, H. F. (2019). Exploring the Role of Insulin Signaling in Relative Growth: A Case Study on Wing-Body Scaling in Lepidoptera. *Integrative and Comparative Biology*, 59(5), 1324–1337. <https://doi.org/10.1093/icb/icz080>

Mirth, C. K., Truman, J. W., & Riddiford, L. M. (2009). The Ecdysone receptor controls the post-critical weight switch to nutrition-independent differentiation in *Drosophila* wing imaginal discs. *Development*, 136(14), 2345–2353. <https://doi.org/10.1242/dev.032672>

Nijhout, H. F. (1999). Control Mechanisms of Polyphenic Development in Insects: In polyphenic development, environmental factors alter some aspects of development in an orderly and predictable way. *BioScience*, 49(3), 181–192. <https://doi.org/10.2307/1313508>

Oliveira, M. M., Shingleton, A. W., & Mirth, C. K. (2014). Coordination of Wing and Whole-Body Development at Developmental Milestones Ensures Robustness against Environmental and Physiological Perturbations. *PLOS Genetics*, 10(6), e1004408. <https://doi.org/10.1371/journal.pgen.1004408>

Pavlopoulos, A., & Akam, M. (2011). Hox gene *Ultrabithorax* regulates distinct sets of target genes at successive stages of *Drosophila* haltere morphogenesis. *Proceedings of the National Academy of Sciences*, 108(7), 2855–2860. <https://doi.org/10.1073/pnas.1015077108>

Weatherbee, S. D., Halder, G., Kim, J., Hudson, A., & Carroll, S. (1998). *Ultrabithorax* regulates genes at several levels of the wing-patterning hierarchy to shape the development of the *Drosophila* haltere. *Genes & Development*, 12(10), 1474–1482. <https://doi.org/10.1101/gad.12.10.1474>

Author's Response to Decision Letter for (RSPB-2021-1605.R0)

See Appendix A.

RSPB-2021-2764.R0

Review form: Reviewer 1 (John Spence)

Recommendation

Major revision is needed (please make suggestions in comments)

Scientific importance: Is the manuscript an original and important contribution to its field?

Excellent

General interest: Is the paper of sufficient general interest?

Good

Quality of the paper: Is the overall quality of the paper suitable?

Excellent

Is the length of the paper justified?

Yes

Should the paper be seen by a specialist statistical reviewer?

No

Do you have any concerns about statistical analyses in this paper? If so, please specify them explicitly in your report.

No

It is a condition of publication that authors make their supporting data, code and materials available - either as supplementary material or hosted in an external repository. Please rate, if applicable, the supporting data on the following criteria.

Is it accessible?

No

Is it clear?

Yes

Is it adequate?

Yes

Do you have any ethical concerns with this paper?

No

Comments to the Author

To begin, I congratulate the authors on producing a much-improved version of this manuscript. I think that this version sets their interesting new findings usefully in a well-informed context of what has been done with this problem earlier and the biology of the focal animal. I've enjoyed reading the manuscript as I am sure will others interested in its general topic. I have no major reservations about the paper, but raise some quibbles under 'Specific Comments' below that the authors may wish to address in producing a revised version for the press. In addition, I offer many suggestions directly on the text that are not amplified in the 11 numbered points below in hopes that the authors may consider them useful. They relate mainly to economy and clarity of expression.

As I did with the first version, I've converted the PDF to a Word document so that I could suggest edits in 'track-changes' that might be useful of the authors. Conversion leaves many glitches and, although I've tried to catch and fix these, I apologize for any that I have not caught.

Specific Comments

1. (Methods) How many bugs and how many times were stocks 'replenished', even roughly? It is not clear how this may have affected composition of the breeding stock. By stating the size of these additions, one can judge whether they could have affected the basal condition of the breeding stock after 7 years in the lab. A simple and reasonable IMO way to deal with this is to lay it bare and if it seems likely, state that you believe that these additions did not substantially affect the stock. I just have no real idea how much 'selection' goes on in 7 years of lab breeding at

16:8, which is longer constant daylength than the bugs ever perceive in the environment that they were taken from. Of course, I always say that it is better to light a candle than to curse the darkness, although in science it is always good to remember what lies beyond our view.

2. (Methods) I assume you actually mean forewings or hemelytra, or did you actually tease out the reduced flight wings?

3. (Methods) Densities may have varied box to box during the experiment to assess density effects. I guess that the 'numbers out' provide some rough and ready assessment of how important this could be. It might be important to assure readers that the output numbers for the 4 density treatments were statistically different and distributed on the gradient of abundance as expected.

4. (Methods) Under the Nutrition experiments and elsewhere I think that the authors must be explicit about photoperiod .. in these experiments, if it is the same as the hatching photoperiod, they could write more simply something like "groups ... were hatched and reared in both photoperiods". These kinds of details are necessary to meet the criterion of repeatability. The way I read it right now is that starting groups of treatments were a random mix of first instars hatched at the two different photoperiods, and thus the experiments are flawed by this uncontrolled factor, although I imagine that this is not the case. Perhaps it is thought that hatching photoperiod does not matter, but if so this needs to be clear. This is also a good place to mention, that I think that somewhere toward the beginning of the paper the authors must explain what they are calling 'long' and 'short' photoperiods and be sure to adopt a consistent terminology and designation throughout the paper. I personally prefer the simple designations like, for example, '16L:8D' because they are unaambiguous, but I think that anything will work as long as it is defined and consistently employed. Also, I think that the relationship between constant photoperiods administered in the lab and natural daylight under which the polyphenism has evolved and is regulated in nature is much better rationalized in this version of the manuscript; however, I would look carefully at this aspect throughout the MS, say, e.g., in Sect. 3.a., to see if further improvement is possible.

5. (Methods) Although things are much improved in this version, I still struggle a little with understanding exactly what was done on the molecular side. It would be useful for readers like me, if the authors could succinctly explain what they were aiming to achieve with the injections. Expressions like 'knockdown' or 'depletion' give some notion that the objective was to nullify the action of particular genes, but can one clearly explain the mechanics of how this is being accomplished. There are a series of comments on the MS in track changes to suggest where more complete discussion could help neophyte readers better understand what was done

6. (Methods) The last sentence of the Methods section is quite uninterpretable to me. For me to understand, more context is required.

7. (Results) I interpret Fig. 2A in a way somewhat different from what is written at this point. Please consider the rewording that I've suggested or clarify what is meant.

8. (Results) It took me a little while to sort out the meaning of the text here. I suppose you mean to refer only to the first instars reared at 18L:D6, given the df associated with the test. Given the low sample for those reared with low diets, I am not sure that a χ^2 is valid, and as you say the results must be treated cautiously. The results may suggest that the 18/6-L combination was developmentally stressful. In Alberta, mesopteroous bug are rare and, in my memory, occur exclusively during the time that the population mode is shifting between SW and LW.

9. (Methods) Were these experiments controlled for the possible simple effects of injection? It seems to me that injection of equal amounts of saline or some other innocuous solution is required to be sure of interpretations.

10. (Discussion) I agree that this is one way to interpret the results. I prefer the idea that there is individual variation in any population for critical stage, so there is always room for selection to move a population with changing climate. In some years, an individual that commits in 3rd instar may lose some or all of its reproductive output because there is an early winter. However, as winters drift later, such individuals would be favoured. I'd bet that individuals have relatively narrow windows for induction although the window may be wide in a population. I may be wrong, but I don't think one can prove the point with what we know and I think the discussion ought to be equivocal about it.

11. (Discussion) How do we know that even this qualified statement is correct? Wing buds begin to appear in instar 3 and are obvious in instar 4. Wings appear at the adult molt and as far as I could see, there is no difference between wing bud length between nymphs that become either micropters or macropers.

Review form: Reviewer 2

Recommendation

Accept with minor revision (please list in comments)

Scientific importance: Is the manuscript an original and important contribution to its field?

Good

General interest: Is the paper of sufficient general interest?

Good

Quality of the paper: Is the overall quality of the paper suitable?

Good

Is the length of the paper justified?

Yes

Should the paper be seen by a specialist statistical reviewer?

Yes

Do you have any concerns about statistical analyses in this paper? If so, please specify them explicitly in your report.

Yes

It is a condition of publication that authors make their supporting data, code and materials available - either as supplementary material or hosted in an external repository. Please rate, if applicable, the supporting data on the following criteria.

Is it accessible?

Yes

Is it clear?

Yes

Is it adequate?

Yes

Do you have any ethical concerns with this paper?

No

Comments to the Author

The authors have done a very good job revising their manuscript. I do have a few minor suggestions, but otherwise the MS looks excellent.

Abstract lines 28-30. This study failed to find genetic pathways underlying wing development. Ultimately, the end of the abstract needs some wider take home message.
line 149. a binomial error structure and logit link function

Because Figure 1 is not included in the manuscript, I failed to assess whether authors' interpretation is correct or not (e.g., lines 236-265; 417-473)

Results: Chi-square values should be quoted to two decimal places.

line 344. Please summarize your experiments more clearly.

lines 354-358. Please add a reference.

Decision letter (RSPB-2021-2764.R0)

14-Feb-2022

Dear Mr Gudmunds:

Your manuscript has now been peer reviewed and the reviews have been assessed by an Associate Editor. The reviewers' comments (not including confidential comments to the Editor) and the comments from the Associate Editor are included at the end of this email for your reference. As you will see, the reviewers and the Editors have raised some concerns with your manuscript and we would like to invite you to revise your manuscript to address them.

Research ethics:

Use of animals and field studies:

It is a condition of publication that you make available the data and research materials supporting the results in the article (<https://royalsociety.org/journals/authors/author-guidelines/#data>). Datasets should be deposited in an appropriate publicly available repository and details of the associated accession number, link or DOI to the datasets must be included in the Data Accessibility section of the article (<https://royalsociety.org/journals/ethics-policies/data-sharing-mining/>). Reference(s) to datasets should also be included in the reference list of the article with DOIs (where available).

If you wish to submit your data to Dryad (<http://datadryad.org/>) and have not already done so you can submit your data via this link [http://datadryad.org/submit?journalID=RSPB&manu=\(Document not available\)](http://datadryad.org/submit?journalID=RSPB&manu=(Document%20not%20available)), which will take you to your unique entry in the Dryad repository.

Please submit a copy of your revised paper within three weeks. If we do not hear from you within this time your manuscript will be rejected. If you are unable to meet this deadline please let us know as soon as possible, as we may be able to grant a short extension.

Best wishes,
Dr Sasha Dall
mailto: proceedingsb@royalsociety.org

Associate Editor Board Member

Comments to Author:

I thank the authors for the very thorough revision of this manuscript and their efforts in conducting additional experiments to strengthen the results around density and nutrition. The revised manuscript has been reviewed by two of the original reviewers and by myself. Both reviewers and I were impressed by the new version of the manuscript; it is a clear presentation of these interesting findings, with a more nuanced discussion.

Both reviewers suggest additional changes, mostly around clarifying (especially) the methods and results, and some additional suggestions for the discussion. As you will see, Reviewer 2 has carefully reviewed the revised manuscript and commented extensively on the word file. The authors need not undertake every change, but I suggest they consider them because many of the suggestions would make the writing crisper and cleaner; more readable.

Please also correct the following typos and check the manuscript for similar typos throughout.

These line numbers correspond to the 'track changes' version

Line 64: delete different

Line 96: 'points'

Line 111: 'seems'

Line 115: 'have'

Line 139: 'were'

Line 145: 'stock populations were' or 'stock population was'

Line 500: 'have' instead of 'has' in both instances

Line 566: change 'by' to 'from'

Line 578: replace the comma with a semi-colon or put the phrase beginning 'see' in parentheses

Line 588: Replace 'E.g.' with 'For example,'

Line 611: I think the authors mean 'relay' instead of 'rely'

Line 617: add 'the' before sensitive

Line 656: 'see an effect, or because' (remove italics on 'or' and delete comma after or)

Line 658: Full stop instead of colon

Line 663: Add 'The' before expression; 'was' rather than 'were'

Line 686: 'controls'; delete 'the' after controls

Line 687: delete 'due to'

Line 735: 'result'

Reviewer(s)' Comments to Author:

Referee: 2

Comments to the Author(s).

The authors have done a very good job revising their manuscript. I do have a few minor suggestions, but otherwise the MS looks excellent.

Abstract lines 28-30. This study failed to find genetic pathways underlying wing development.

Ultimately, the end of the abstract needs some wider take home message.

line 149. a binomial error structure and logit link function

Because Figure 1 is not included in the manuscript, I failed to assess whether authors' interpretation is correct or not (e.g., lines 236-265; 417-473)

Results: Chi-square values should be quoted to two decimal places.

line 344. Please summarize your experiments more clearly.

lines 354-358. Please add a reference.

Referee: 1

Comments to the Author(s).

To begin, I congratulate the authors on producing a much-improved version of this manuscript. I think that this version sets their interesting new findings usefully in a well-informed context of what has been done with this problem earlier and the biology of the focal animal. I've enjoyed reading the manuscript as I am sure will others interested in its general topic. I have no major reservations about the paper, but raise some quibbles under 'Specific Comments' below that the authors may wish to address in producing a revised version for the press. In addition, I offer many suggestions directly on the text that are not amplified in the 11 numbered points below in hopes that the authors may consider them useful. They relate mainly to economy and clarity of expression.

As I did with the first version, I've converted the PDF to a Word document so that I could suggest edits in 'track-changes' that might be useful of the authors. Conversion leaves many glitches and, although I've tried to catch and fix these, I apologize for any that I have not caught.

Specific Comments

1. (Methods) How many bugs and how many times were stocks 'replenished', even roughly? It is not clear how this may have affected composition of the breeding stock. By stating the size of these additions, one can judge whether they could have affected the basal condition of the breeding stock after 7 years in the lab. A simple and reasonable IMO way to deal with this is to lay it bare and if it seems likely, state that you believe that these additions did not substantially affect the stock. I just have no real idea how much 'selection' goes on in 7 years of lab breeding at 16:8, which is longer constant daylength than the bugs ever perceive in the environment that they were taken from. Of course, I always say that it is better to light a candle than to curse the darkness, although in science it is always good to remember what lies beyond our view.
2. (Methods) I assume you actually mean forewings or hemelytra, or did you actually tease out the reduced flight wings?
3. (Methods) Densities may have varied box to box during the experiment to assess density effects. I guess that the 'numbers out' provide some rough and ready assessment of how important this could be. It might be important to assure readers that the output numbers for the 4 density treatments were statistically different and distributed on the gradient of abundance as expected.
4. (Methods) Under the Nutrition experiments and elsewhere I think that the authors must be explicit about photoperiod .. in these experiments, if it is the same as the hatching photoperiod, they could write more simply something like "groups ... were hatched and reared in both photoperiods". These kinds of details are necessary to meet the criterion of repeatability. The way I read it right now is that starting groups of treatments were a random mix of first instars hatched at the two different photoperiods, and thus the experiments are flawed by this uncontrolled factor, although I imagine that this is not the case. Perhaps it is thought that hatching photoperiod does not matter, but if so this needs to be clear. This is also a good place to mention, that I think that somewhere toward the beginning of the paper the authors must explain what they are calling 'long' and 'short' photoperiods and be sure to adopt a consistent terminology and designation throughout the paper. I personally prefer the simple designations like, for example, '16L:8D' because they are unambiguous, but I think that anything will work as long as it is defined and consistently employed. Also, I think that the relationship between constant photoperiods administered in the lab and natural daylight under which the polyphenism has evolved and is regulated in nature is much better rationalized in this version of the manuscript; however, I would look carefully at this aspect throughout the MS, say, e.g., in Sect. 3.a., to see if further improvement is possible.
5. (Methods) Although things are much improved in this version, I still struggle a little with understanding exactly what was done on the molecular side. It would be useful for readers like me, if the authors could succinctly explain what they were aiming to achieve with the injections. Expressions like 'knockdown' or 'depletion' give some notion that the objective was to nullify the action of particular genes, but can one clearly explain the mechanics of how this is being accomplished. There are a series of comments on the MS in track changes to suggest where more complete discussion could help neophyte readers better understand what was done
6. (Methods) The last sentence of the Methods section is quite uninterpretable to me. For me to understand, more context is required.
7. (Results) I interpret Fig. 2A in a way somewhat different from what is written at this point. Please consider the rewording that I've suggested or clarify what is meant.
8. (Results) It took me a little while to sort out the meaning of the text here. I suppose you mean to refer only to the first instars reared at 18L:D6, given the df associated with the test. Given the low sample for those reared with low diets, I am not sure that a χ^2 is valid, and as you say the results must be treated cautiously. The results may suggest that the 18/6-L combination was developmentally stressful. In Alberta, mesopterous bug are rare and, in my memory, occur exclusively during the time that the population mode is shifting between SW and LW.

9. (Methods) Were these experiments controlled for the possible simple effects of injection? It seems to me that injection of equal amounts of saline or some other innocuous solution is required to be sure of interpretations.

10. (Discussion) I agree that this is one way to interpret the results. I prefer the idea that there is individual variation in any population for critical stage, so there is always room for selection to move a population with changing climate. In some years, an individual that commits in 3rd instar may lose some or all of its reproductive output because there is an early winter. However, as winters drift later, such individuals would be favoured. I'd bet that individuals have relatively narrow windows for induction although the window may be wide in a population. I may be wrong, but I don't think one can prove the point with what we know and I think the discussion ought to be equivocal about it.

11. (Discussion) How do we know that even this qualified statement is correct? Wing buds begin to appear in instar 3 and are obvious in instar 4. Wings appear at the adult molt and as far as I could see, there is no difference between wing bud length between nymphs that become either micropters or macropters.

Author's Response to Decision Letter for (RSPB-2021-2764.R0)

See Appendix B.

Decision letter (RSPB-2021-2764.R1)

25-Mar-2022

Dear Mr Gudmunds

I am pleased to inform you that your manuscript entitled "Photoperiod controls wing polyphenism in a water strider independently of insulin receptor signaling" has been accepted for publication in Proceedings B.

Data Accessibility section

Open Access

Paper charges

Sincerely,

Dr Sasha Dall

Associate Editor:

Board Member

Comments to Author:

I thank the authors for their careful revision of the manuscript. I look forward to seeing this interesting study published.

Appendix A

UPPSALA
UNIVERSITET

Uppsala 15.12.2021

Dear editor,

Please find resubmission for manuscript RSPB-2021-1605 entitled: “Photoperiod controls wing polyphenism in a water strider independently of insulin receptor signaling” for potential publication in PRSB attached.

We are very grateful to the three reviewers and the associate editor for their helpful comments and thoughts on the manuscript and we have now significantly revised our manuscript, including repeating and expanding our original photoperiod, density and nutrition experiments.

More specifically we have:

- 1) Revised our introduction and discussion to accommodate more literature on wing polymorphisms and the developmental trajectory as suggested
- 2) Repeated density and nutrition experiments with increased sample size and re-written the material and methods section for this part
- 3) Demonstrate that the low nutrition treatment leads to lower survival, longer developmental duration and smaller body size (as asked for by reviewer 2)
- 4) Revised the text throughout and included most of the references suggested by the reviewers

In particular the increased sample size for the nutrition and density treatments we believe has significantly increased the strength of the inferences in our manuscript because although results are entirely consistent with our earlier findings, they are now much more robust.

Detailed responses to editor and reviewers’ comments can be found below with our response below the reviewer comments.

Please do not hesitate to contact us if you have any questions and we look forward to hear from you.

Best wishes,

Erik Gudmunds (on behalf of all authors)

Responses to referee comments for the manuscript: "Photoperiod controls wing polyphenism in a water strider independently of insulin receptor signaling components".

We have put comments from editor and referees in bold text and our responses in normal font. Changes in the manuscript are written in italic font.

Associate Editor

Board Member: 1

Comments to Author:

The manuscript under consideration was reviewed by three experts and myself. The study tackles the question of environmental factors mediating wing polyphenism in a water strider, through laboratory experiments manipulating temperature, density and nutrition, and using RNA interference to test the role of insulin signaling. All reviewers found the work interesting, as did I. All reviewers raised important points that I believe should be addressed.

Reviewer 1 suggests an alternative model for wing development: that all individuals begin on a long-winged trajectory and some switch to a short-winged trajectory depending on environmental conditions. This model should be explicitly considered, along with the strength of evidence for whether it's the length of photoperiod in the critical developmental window that matters, or whether photoperiod is increasing or decreasing.

We have considered this model in the discussion (L403-423). Also, we discuss the matter of increasing/decreasing photoperiod or photoperiod length in light of our data and the data published in Spence (1989), L352-371.

Reviewer 1 also suggests stronger links between the manuscript's hypotheses and the literature, including linking hypotheses about wing development to reproductive strategy. Both reviewers 1 and 2 suggest relevant literature that should be cited.

We now cite and discuss the relevant literature that was suggested by reviewer 1 and 2. Regarding the relationship between wing morph and reproductive strategy, we cite several important works on the subject, but due to space limitations and because we did not investigate reproductive strategy *per se* in the manuscript, we do not discuss this complex relationship explicitly.

The reviewers have raised important points about the methods that need attention and discussion. Reviewer 1 points out that the photoperiod range tested is greater than natural variation and that a sudden switch is artificial, and Reviewer 2 points out that the short photoperiod appears to be a more extreme treatment than the long photoperiod.

We acknowledge that the M&M-section was unclear and that the degree of change in photoperiod used for the switch experiment is artificial. The reason we have chosen these specific photoperiods is of practical concerns because we are dependent on having information about which morph individuals will develop into in order to know the potential effect of RNAi treatment and qPCR results. We have added several notes about this issue both in the M&M and the Discussion.

L112-115. "In relation to the photoperiod conditions in the area the population of G. buenoi originate from, the 14:10, 15:9 and 16:8 photoperiods covers the range a developing nymph is exposed to (Toronto solstice daylength is 15:27 hh:mm, 16:39 including civil twilight)."

L142-147. "A caveat with this shifting experiment is that the change in photoperiod (from 12:12 to 18:6 or vice versa, resulting in a 6-hour shift) is artificial and thus our results might not reflect the same dynamics of wing morph induction in natural conditions. Furthermore, these two photoperiods are outside the range of photoperiods that the native Toronto G. buenoi population likely can be exposed to during nymphal development."

L412-414. "It should be noted that these results may be specific for the tested conditions, especially considering that 12:12 and 18:6 L:D and the six hour shifts in photoperiod is extreme and not experienced by individuals in natural conditions."

Reviewer 1 also notes concerns about the density treatment, in that the low-density treatment is an isolation treatment; the reader also need to see the results from low, medium and high treatments separately.

To accommodate these concerns, we have re-performed the density experiment in order to have better control of starting conditions and to explore the effects of density in both photoperiods (previously only in 18:6). We believe our description of the new density experiment in the M&M, Results and Discussion should be sufficient to accommodate the concerns raised by Reviewer 1.

It would be helpful if the authors can provide evidence that the low nutritional treatment was stressful enough to provoke a response to the treatment, as Reviewer 2 suggests.

We have re-performed the nutrition experiments in 12:12 and 18:6 and recorded developmental duration, mortality and differences in adult body size. We believe that this will show clearly that restrictions in nutritional availability is stressful for individuals and that it provokes a clear response on developmental duration, body size and mortality.

Reviewers 1 and 2 ask for clarification about important aspects of the methods, especially the density treatment and the feeding schedule and severity of the low-food treatment, which affects interpretation of the nutrition treatment.

See above. Also, the structure of the M&M has been revised for the sake of clarity by separating the description of each experiment with a sub-heading.

Response to referee #1:

We are grateful to you for taking the time to review our manuscript thoroughly and that you kindly suggested specific changes in the manuscript itself. Please find below our response to your comments.

I found this work of much interest, and think that it can be turned into a useful, and likely more easily understood, publication about wing-dimorphism in *Gerris buenoi*. Having done some work with this wonderful species, I agree that this *G. buenoi* could be a most appropriate model system for exploring wing polyphenisms in semi-aquatic bugs in a way that can likely improve general understanding of of this life-history phenomenon. In the attached Word file edited in ‘track changes’, I offer suggestions about many specific points that I think merit attention in improving this paper.

Thank you for this reassuring comment and the suggestions of specific changes directly in the text.

I maintain that it is important, in this sort of work, to choose one’s photoperiods in clear relation to the source of the population that one is working with. Thus, it is not clear to me why the two photoperiods employed for use in these experiments were chosen, or how this choice helps the argument that photoperiod actually drives induction of wing-morph in nature. I believe that photoperiod is most important, but fail to see how the experiments here provide definitive proof more compelling than other work. Here’s the problem. Maximum photoperiod (daylight+civil twilight) as solstice in Toronto, Canada is 16h 39m. Minimum photoperiod that 3rd instars are ever likely to see (based on a highly conservative guess of 15 April) is 14h 12m of daylight, and on 20 Sept, likely as late as one could ever find a 3rd or 4th instar on a pond in southern Ontario daylength is about 13h 30m.

We fully agree with your notion that the two photoperiods 12:12 and 18:6 L:D are outside the natural photoperiodic range of the source population. We chose these two extreme photoperiods as we wanted to find conditions which reliably could induce ~100% of each wing morph, with the reason to facilitate the functional genetic work (both as reported in the manuscript but also for other experiments). To accommodate the issues with the range of photoperiod you raised, and its impact on the inferences we can make about the morph-inducing characteristics of photoperiod in the source population, we have modified several sentences (see below). We also like to emphasize that in the constant photoperiod experiments (Figure 1) we explored a range of photoperiods that are relevant in regard to what the source population has been exposed to.

*L142-147. “A caveat with this shifting experiment is that the change in photoperiod (from 12:12 to 18:6 or vice versa, resulting in a 6-hour shift) is artificial and thus our results might not reflect the same dynamics of wing morph induction in natural conditions. Furthermore, these two photoperiods are outside the range of photoperiods that the native Toronto *G. buenoi* population likely can be exposed to during nymphal development.”*

*L112-115. “In relation to the photoperiod conditions in the area the population of *G. buenoi* originate from, the 14:10, 15:9 and 16:8 photoperiods covers the range a developing nymph is exposed to (Toronto solstice daylength is 15:27 hh:mm, 16:39 including civil twilight).”*

L412-414. “It should be noted that these results may be specific for the tested conditions, especially considering that 12:12 and 18:6 L:D and the six hour shifts in photoperiod is extreme and not experienced by individuals in natural conditions.”

Essentially, all of the induction action that is of any significance to these bugs in nature happens over a range of 2.5-3.0 hrs of daylight, rather than as a consequence of the sudden 6 h shifts that they were subjected to in the experiments reported in this paper. In the experiments reported in this paper, two photoperiods 18:6 and 12:12 both with daylengths, respectively, longer and dramatically shorter than will be ever relevant to natural populations of *G. buenoi* in southern Ontario.

Indeed, the six-hour shifts are, as you point out, very sudden and unnatural. We have modified the M&M and Discussion to explicitly state that this can impact interpretation of the results (see response to comment above).

Furthermore, the way I read it, stock cultures were held at 20:2 for some reason not explained, and it is not stated how long (or for how many generations) they were held under such conditions. Please consider the selection that may have been imposed, and how it might have affected responses that you seek to illuminate.

We have revised the description of keeping of the stock population in the M&M.

*L86-96. "The *G. buenoi* population was originally collected in a pond in Toronto, Ontario, Canada, during 2012, but has been replenished several times since then from populations in the same area. The photoperiods for the stock population was from 2012 to May 2019 16:8 light:dark (L:D) and since then 22:2. From May 2019 until present the generation of adults in the breeding stock was primarily from individuals that had been reared during their nymphal stages in either 12:12 or 18:6 L:D, constituting a mix of short-winged (henceforth micropterous) and long-winged (henceforth macropterous) morphs. A smaller proportion of replenishment came from nymphs reared at high densities in 22:2 which produces a small frequency of macropterous individuals. The stock population were fed five times a week with frozen crickets (*Acheta domestica*) and kept at room temperature. Pieces of Styrofoam were provided as a substrate for egg laying and resting."*

Also, just to raise one last criticism, which I understand can be levied at most of us who do lab work with this business, is that induction brought about by sudden and large shifts (i.e., >2x the full range of photoperiod difference ever seen in nature) in photoperiod may have little to do with how induction actually happens in nature. *Under such conditions, it is possible that many links in the process could go missing because they just were not initiated.*

Therefore, although statements about how photoperiod switches might function in relation to LD and SD regimes are reasonable, these findings must be quite cautiously interpreted with respect to understanding mechanisms that operate in nature. This is the reason why in my own fundamental work on these questions (Spence 1989), I worked with enclosed natural populations and focused mainly on understanding critical stages in relation to natural photoperiods in relation to manipulations that I could accomplish. I submit that this provides a much more reasonable basis for inferences about the clear importance of photoperiod in wing-morph induction than does work involving sudden shifts between two constant photoperiods well outside the range of daylengths that the bugs would ever encounter in nature.

We agree and have taken this into consideration by modifying a number of sections in the M&M and Discussion. For specific changes, see responses to comments above.

The important findings wrt to the photoperiod issue in the 1989 study were: 1) 100% of all bugs that reached the imago stage before the solstice, either male or female, were SW; 2) nymphs that were in instars 3-5 (+1 2nd instar) at the solstice gave rise to an increasing proportion of SW imagoes; 3) females were more likely than males to become SW adults in all categories (very interesting adaptive significance in my mind – in fact, one must wonder why males ever give up flight?); and 4) (fortunately) wrt to Alberta *G. buenoi* anyway, the same basic results are obtained whether the induction happens under daily (increasing or decreasing) shifts in photoperiod or under constant photoperiods in the lab. Thus, at least for *G. buenoi* in mid-northern latitudes, one need not be distracted by Kari Vepsäläinen’s interesting finding that direction of photoperiod change seems to matter in some Finnish species.

Thank you for making us aware of this information. We have specifically added information about the results from this publication in the Discussion and have also added a reflection on whether photoperiod change is an important cue for *G. buenoi* compared to the sister species *G. odontogaster*. Important changes are found in L352-370.

What seemed most significant to me from the 1989 paper is that there is clearly variation in sensitive stage. I suppose that this variation has genetic basis and is subject to selection. I believe that I have shown that it is, albeit in (unfortunately) unpublished work. We were able to select for \pm complete expression of either the LW or SW condition under 19:5 (the longest natural photoperiod in Edmonton) in the laboratory using mass cultures. As I recall we got there in 12-15 generations and ran the cultures for >20 lab generations. Now that I have shed my administrative load, retired, and survived and recovered from two surgeries, I hope to get on with putting this work into print. At any rate, the point I am trying to make here and which could be helpful to for the authors, is that considering the critical stage for induction of both wing-morph and breeding status as quite variable allows us to imagine how this species can adjust its life history in time and space.

That is very interesting, thank you for sharing the results of this unpublished work. Considering the results from Spence (1989) we have re-evaluated several aspects of the results in our photoperiod shift experiment. Specifically, we have added a note regarding the variation observed within experimental groups in both shifting directions, and suggest that it could be explained by genetic variation in sensitivity to photoperiod (constant or changes in photoperiod).

L420-423. “Interestingly, there is variation in wing morphs within some experimental groups (e.g. I5E and I5M in the 12:12 to 18:6 shift, Figure 1E), which may be caused by genetic variation in sensitive stage for induction.”

Given such variation, I am quite suspicious of the conclusion in the paper that results show significant asymmetry in induction depending on whether SW-LW or LW-SW transitions are considered. In fact, as explained below (point #3), I think this is rather

wonng-headed. Although I understand that your paper has focused more on the mechanism of wing-length (btw \neq 'wing size') determination, any discussion or speculation about adaptive significance depends on understanding aspects of induction related to its timing.

The individuals in the experiment are randomly sampled from the stock population and thus we find it unlikely that the asymmetric response has its basis in genetic variation. However, as above, the variation within experimental groups could have its source in genetic variation for threshold or sensitive stage.

Linkage between breeding status and voltinism. I was quite surprised to find no mention of connections between wing-length and breeding/diapause status in this paper, as this has been a centerpiece of previous work to set gerrid polyphenisms in the context of adaptive strategies. These features seem to be part of an 'adaptive syndrome' similar, but not quite the same, as C. G. Johnson's classic oogenesis-flight syndrome. Clearly, from previous work, it is clear that SW bugs are virtually all direct breeders in *G. buenoi*. In more than 20 years of serious work with this species, and in handling thousands of overwintered individuals, I have only ever encountered ONE SW individual among overwintered populations! The situation for LW bugs is more interesting because some breed directly while others must traverse and overwintering diapause before breeding in the next year.

We agree that these are very interesting findings, but it is not something we have focused on in this work and space limitations limit our discussion to the morph-determining factors.

I am guessing that all bugs that developed under 20:2 in the stock cultures were direct breeders. Were they? Is there any way to know based on records from early cultures?

Yes, individuals raised in 18:6 or 22:2 were direct breeders, whereas individuals from 12:12 does not breed until ~2 weeks after being transferred to 22:2 (the stock culture room).

What was the proportion of SW adults in these source cultures? The argument that the mix of morphs resulted from some conscious choice of rearing density seems dubious to me,

We have repeated the density experiments and find high density to increase proportion of macropterous individuals. While we have not measured the proportion SW/LW in the stock population, variation in rearing density could contribute to variation in frequency of long and short winged individuals. The description of the stock population has been revised in the M&M (see above).

and I emphasize that the results presented in this paper are at odds with those presented by Harada & Spence (2000). I do not argue that one set is right and the other wrong, but rather that in order to really understand density effects we need to understand why these different results were achieved. Honestly, I personally like the finding reported in this paper that high density leads to more LW bugs better than the contrary proposition, which is what we found, but I just cannot understand the differences. I just cannot calculate a comparison of densities used because clear information is not provided for high density treatments in this paper. Presuming that

you report starting densities, our low-density treatments were roughly equivalent to yours, but perhaps our high-density treatments were somewhat more dense than yours. However, if high density promotes development of the LW state, one would imagine that we'd have seen different results in Edmonton. Also, as a tangent here, although it is expressed in the track changes, I must point out that having one bug in a box, no matter what the size of the box, is not a density equivalent to that which includes interactions with other individuals. Social interactions are doubtlessly crucial to the sorts of density effects that one hopes to elucidate with such experiments.

To address this comment we have now repeated the density experiment. In the revised experiment we did not include isolation as a treatment, since we agree that it not a density-equivalent. We now specifically mention that the results presented in our paper contrasts that of Harada & Spence (2000).

L384-394. "In line with several other studies on Gerromorpha (23,25,37) we found that higher density led to increased frequency of macropterous individuals (Fig 2). This is in contrast to results by Harada and Spence (24) who found that high density increases the frequency of micropterous G. buenoi individuals and it is not immediately clear why, although it is possible that the different photoperiods used in our experiment and that by Harada and Spence (24) could play a role. Increased macropter frequencies in response to high nymphal density is likely an adaptive response to enable dispersal when conditions may become detrimental in the future, e.g. due to competition for food. However, the induction of micropters in response to crowding may emphasize that the morph induction can be rather complex, see (24) for a more comprehensive discussion on life-history strategies in response to crowding conditions in water striders."

The model for adaptive significance. As I digested the paper, it seemed to me that the authors envisaged an altogether different model than I do or that has been employed by others. The discussion and reasoning employed in the paper presupposes that nymphal development is affected by ambient photoperiod from the time or hatching – or perhaps even earlier.

With this model in mind, it is logical to expect 'switching' to affect wing development either way. This is not in line with previous thinking, although a rationale for contrast was not presented. Based on the work of Nils Moller Andersen, Kari Vepsäläinen (and its students), me and virtually all others who have worked on this problem, I see it quite differently.

We regret that the model for wing morph determination in the context of mechanisms underlying polyphenisms was not explained better because we agree with the reviewer. We have now revised the text to better explain our thinking, and we have re-evaluated the results of the shift experiment to one which makes less inferences to natural induction.

L403-423. "From the experiments with constant photoperiod we show that exposure to a long photoperiod throughout development resulted in micropterous development, whereas exposure to a short photoperiod resulted in macropterous development. To explore the sensitive stages of wing morph determination in G. buenoi we exposed nymphs to a shift in photoperiod at different developmental stages, similar to an experimental design reported by Inoue & Harada (28) with A. paludum. Our results suggest that exposure to a long photoperiod (18:6 L:D) is highly inductive of the micropterous morph and that the sensitive

window for this induction lasts from at least instar 3 until latest day 2 in instar 5. Furthermore, our results show that the growth of the adult wings, which occurs in instar 5, can by some mechanism be stunted by photoperiod cues that are received as early as instar 3. It should be noted that these results may be specific for the tested conditions, especially considering that 12:12 and 18:6 L:D and the six hour shifts in photoperiod is extreme and not experienced by individuals in natural conditions. Nevertheless, these experiments provide valuable information on the potential limits of induction, including the responsiveness of the mechanism that rely information of photoperiod to the growing adult wing tissue. We suggest that the pattern of commitment to wing morph we observe is in line with a view that the default developmental trajectory is towards the macropterous morph, but that this development can be arrested by an unknown mechanism during a rather long sensitive window, and lead to micropterous development.”

I note that all individual *G. buenoi*, whether they go onto eclose as SW or LW development, show development of external wing buds. These, I thought, might be detectable in instar 2 but they are clearly visible as slight swelling in most instar 3 nymphs and, of course, are dead obvious in all instar 4 and 5 nymphs. Despite a semi-serious effort I could never see differences in wing-pad development, even in instar 5, between bugs that went on to become either SW or LW imagoes. Thus, early lives of all developing nymphs include some developmental progress toward developing wings. IMO, there is, for each individual (with variation in the population), one critical period during development, during which the investment in full-wings and at least some of the associated machinery can be turned off. If it is not turned off at that time, normal wing and muscle development continues.

Given my understanding, which I think is very well supported by previous work in both *G. buenoi* and other gerrid species, I think it is simply ‘wrong-headed’ to think of bugs initially reared under short-day photoperiods as being programmed to be LW and those reared in early stages under long-day photoperiod to be programmed to be SW. I submit that there is one two-way switch that operates on bugs that are all developing toward being LW adults, and the setting of this switch may be changed by photoperiod experienced at a particular critical stage, the timing of which varies among individuals. There are no nymphs set to develop as SW adults in the early stages of post-embryonic development. IMO, this is not a semantic argument. The paper, as written, rather directly assumes a different chain of events. I believe that this contradicts the best available information, and that if the authors, want to use that assumption anyway in the face of long-established thinking, that they must directly say why and show that the assumption is valid. It took me awhile to understand what was bothering me about the analysis that was presented in the paper, but I have finally understood that this was a most bothersome aspect.

As above, we note that our thinking was not expressed well-enough and could be more in line with previous evidence. We have taken your comment into account by changing and revising a number of aspects on the reasoning of the shifting experiment (see above).

I will emphasize one last point here that relates to an error in your paper. You describe *G. buenoi* as being bi-voltine in Canada. This may be true for a few of the southern-most populations, but even there I am doubtful. It has certainly not been studied. Here in Alberta, I can say with certainty that *G. buenoi* is partially bi-voltine, i.e., overwintered females split their reproductive effort between direct breeding and diapause-bound

progeny. This interesting and critical aspect of life-history is regulated by the processes discussed above.

Thank you for this note. The section where this was written has been removed to accommodate space for a more clear description of the methods.

What is most critically missing in the manuscript is a clear explanation of the methods used that is well integrated with a clear statement of the hypotheses being addressed specifically by each thrust of the research. Also, I personally, struggled with the use of molecular biology methods here that were not explained, but simply described as jargon. If the authors want readers who are interested this work to generally understand what was done, I think that it's rationale and execution must be more clearly explained. I was left in the position of having to simply accept the interesting conclusions as assertions without really understanding their basis. I've tried to help with specific suggestions about how things could be improved in the MS edited in tracked-changes. I think this should not pose much of a challenge to the authors, but that such revision will considerably improve how much the work can be appreciated by readers.

See above for matters regarding explanation of the photoperiod, density and nutrition M&M sections. Regarding the molecular biology work (both in M&M and Results), we have tried our best to explain the reasoning of each step.

Literature that might help. I found that 3 papers quite germane, and I think critical to proper understanding of this work, were missing from the Literature Cited. These are as follows. Two are from my own work dealing directly with *G. buenoi*, and one is a critical foundational paper by Nils Møller Andersen. Although the latter paper is about a couple of Danish species it was amongn the first to lay a clear foundation for appreciating the existence of wing-polyphenism in gerrids. If you have difficulty in finding these papers, I will happily send pdfs of them in response to request to jrspen@gmail.com.

Andersen N.M. 1973: Seasonal polymorphism and developmental changes in organs of flight and reproduction in bivoltine pondskaters (Hem.: Gerridae). Entomol. Scand. 4: 1-20.

Spence, JR. 2000. Seasonal aspects of flight in water striders (Hemiptera: Gerridae). Entomological Science 3: 399-417.

Harada, T & JR Spence. 2000. Nymphal density and life histories of two water striders (Hemiptera: Gerridae). The Canadian Entomologist 132: 353-364.

Thank you for the help in obtaining the references, all papers that were suggested have been cited and their information incorporated in our presentation and reasoning.

Response to referee #2:

Thank you for your time reading and providing comments on our manuscript. Please see below our response to your comments.

Major comments

L94-98 and others. I had the feeling that authors need to introduce a considerable existing literature on the relationship between environmental factors (density, sex ratio, photoperiod, temperature, sex ratio etc) and wing development in water striders. In particular, Tetsuo Harada did lots of works on this issue. In addition, since there are quite a few studies on the genetic basis of wing development in water striders, authors should have discussed them in their manuscript.

We have edited the introduction to include several more references on the environmental factors that have been shown to influence wing morph determination in water striders. We regret that we could not introduce them more in detail, but raise some specific results in the discussion to compare to our results.

L70-72. “The environmental factors influencing wing morph determination has been investigated in a number of water striders and include photoperiod (either constant or gradually changing), temperature and nymphal rearing densities (16,19,22–26).”

*L376-383. “A female bias in morph induction was also present in response to high density (Figure 2A). Interestingly, in a recent publication Han (25) showed that wing morph induction in the Gerrid *Tenagogerris euphrosyne* is sex-biased, where females were equally likely to become macropters as males in low density but in high density females were less likely than males to become macropterous. Taken together, these results suggest that the threshold for induction of wing morphs in water striders can be sexually dimorphic and that the direction of the response can differ between species. For a detailed discussion on the adaptive significance of sex-specific wing morph determination see (25).”*

*L405-408. “To explore the sensitive stages of wing morph determination in *G. buenoi* we exposed nymphs to a shift in photoperiod at different developmental stages, similar to an experimental design reported by Inoue & Harada (28) with *A. paludum*.”*

L220-221, L228-232, L327-330, L339-341. I think that different trajectories (Figure 2) and density effects (Figure 3a) might be because 18:6 is not a “very” strong photoperiod to induce the expression of single wing morph compared to 12:12. In Figure 1b, 12:12 is a very strong photoperiod to induce the expression of single wing morph (LW) because only LW individuals also developed under 14:10. 18:6 might also be a strong photoperiod to induce the expression of single wing morph (SW). However, because a few LW individuals developed under 16:8 condition, 18:6 might not be a “very” strong photoperiod to induce the expression of single wing morph. If authors estimated switcher probabilities across stages under 20:4, they might have got a pattern similar to the one shown in Figure 2c. I think authors need to discuss this possibility. (However, in my personal opinion, I am also in line with authors’ argument that photoperiod is stronger factor to shape gerrid wing polymorphism.)

Also, if authors measured macroptery frequencies across different density conditions under 20:4, they might have found no LW individuals. But, they got LW individuals from the stock population reared under 20:4 and high density, supporting that photoperiod is stronger factor to shape gerrid wing polymorphism.

This is a good point and we now mention this possibility in the discussion.

L412-414. "It should be noted that these results may be specific for the tested conditions, especially considering that 12:12 and 18:6 L:D and the six hour shifts in photoperiod is extreme and not experienced by individuals in natural conditions."

Minor comments

L20-21. Please add why nutrient sensing pathway is important in the development of wing polymorphism.

We modified the sentence to the following to emphasize the potential importance of nutrient sensing in wing polymorphism:

L18-21. "Insect wing polyphenism has evolved as an adaptation to changing environments and a growing body of research suggests that the nutrient sensing insulin receptor signaling pathway is a hot spot for the evolution of polyphenisms, as it provides a direct link between increased growth and the available nutrients in the environment."

L94-98. "Some species" show up a couple of times in this paragraph.

We have re-written this sentence as below to make it easier to read:

L64-67. "Several species are monomorphic for long wings, short wings or are apterous, while some display seasonal wing polyphenism and in yet others wing morph variation seem to be due to genetic polymorphism (18,19)."

L117-122. Please be more specific about how stock populations are maintained. Experiments were done in Sweden using gerrid populations from Canada. How did they maintain genetic diversity of the stock population?

We have added information about the stock population.

*L86-96. "The G. buenoi population was originally collected in a pond in Toronto, Ontario, Canada, during 2012, but has been replenished several times since then from populations in the same area. The photoperiods for the stock population was from 2012 to May 2019 16:8 light:dark (L:D) and since then 22:2. From May 2019 until present the generation of adults in the breeding stock was primarily from individuals that had been reared during their nymphal stages in either 12:12 or 18:6 L:D, constituting a mix of short-winged (henceforth micropterous) and long-winged (henceforth macropterous) morphs. A smaller proportion of replenishment came from nymphs reared at high densities in 22:2 which produces a small frequency of macropterous individuals. The stock population were fed five times a week with frozen crickets (*Acheta domestica*) and kept at room temperature. Pieces of Styrofoam were provided as a substrate for egg laying and resting."*

L119. I missed any explanation of why authors maintained their stock population under such extreme photoperiod condition (20:4)?

See above regarding updated description of keeping the stock population. The reason for keeping them at 22:2 was to accommodate the need of water strider populations from Sweden.

L129-130. I failed to find how authors set starting photoperiods. 18:6 and 12:12? How were photoperiods switched at some developmental stage? Did you increase/decrease photoperiod gradually? or abruptly?

The choice of the photoperiods has been explained and the shifting experiment has been clarified.

L118-123. "To investigate the dynamics of wing morph determination in G. buenoi we performed an experiment in which we exposed nymphs to one starting photoperiod and at specific stages during development we shifted them (without gradual change) to a different photoperiod. A similar experimental design has been used on Aquarius paludum (28). We chose 12:12 and 18:6 L:D as photoperiods, as these were highly predictive for wing morph when exposed to nymphs throughout the entire period of development (the experiments described above)."

L136, 140, 229 and others. I found that there were lots of places citing wrong figures. Please correct them.

Thank you for noting this, we have corrected these errors.

L145-6. How many individuals were kept in high density treatment?

The density experiment has been re-performed in order to better control conditions. See L155-169 for description of the new experiment.

L202-3. I think the appropriate way to show this is to make figure 1b for males and females, respectively. If you revise Figure 1b to show sex specific patterns, I think figure 1c is not necessary.

Good point, we have added sex-specific information about this in the figure. To accommodate space, we also removed the information of parental wing morph in the 15:9 L:D treatment (now in Supplement). Furthermore, we removed figure 1C and instead added a figure on ontogeny of wing buds to accommodate requests from other referees. We also put the shift experiment results in figure 1.

L212-3. Photoperiod increase was from 12:12 to 18:6, and photoperiod decrease was from 18:6 to 12:12?

Yes, this has been clarified in the methods (see response to comment above) and figure legend.

L229-230. I missed the test of interactive effects of density and photoperiod on the expression of wing polymorphism. Why did the authors test density effects only at 18:6 photoperiod condition?

The density experiment has been re-performed in order to better control conditions. See comment above.

L234. Did the food conditions (3,5,7 crickets) affect the developmental duration of nymphs? Because morphological differences between food treatments in Figure 3c were significant but not so dramatic, food deficiency might not be so stressful for gerrids to develop LW.

We have re-performed the nutrition availability experiment and explored differences in mortality, developmental duration, wing morph frequencies and adult size as suggested. See L171-191 and Figure 2 for description of these experiments.

L243. I suggest defining what the low and high diet treatments are.

We have defined this in the “Nutrition”-section in M&M and in the figure legend.

L235-7. Why did you test diet effects only at the extreme photoperiod conditions? I missed diet effects under other photoperiod conditions. Although authors suggested that photoperiod effects were greater than diet effects, the results might be because diet treatments were not so stressful, compared to the extreme photoperiod conditions.

The reason was that we only had access to the rooms in which we could have 14:10, 15:9 and 16:8 for a short while. We acknowledge that a potential effect of diet on wing morph frequencies might be more pronounced if experienced in a less extreme photoperiod. We have added a note that these results may be specific for 12:12 and 18:6.

L395-397. “In the nutrition experiments we exposed individuals to restricted nutrition regimes of different magnitudes and found that the availability of food is not a direct factor for wing morph determination, at least not in the photoperiods we used (Figure 2B&C).”

L335. What do phases mean here?

It means phases in the development of individuals, these are loosely defined and can potentially overlap. We have removed this sentence.

Figure 1b. Please add parental wing morphs for 12:12, 14:10, 16:8 and 18:6 treatments.

We have added information about this in the in the M&M section. Specifically, it was only in 15:9 that we had complete control of parental wing morphs, whereas in other experiments the parental generation was a mix of both morphs (stock population).

L101-104. “Eggs were collected from the stock culture which had adults showing a mix of wing morphs or from controlled crosses of each wing morph (only in the case of 15:9) and randomly distributed into climate-controlled rooms with either 12:12, 14:10, 15:9, 16:8 and 18:6 L:D with ~80 μ Einstein (about 9400 lux) light intensity conditions, at 25°C constant temperature.”

Figure 2a. I understood what the abbreviations mean on the x-axis when I read methods. Please add what the abbreviations mean to the legend.

We have added this information in the new Figure 1 legend.

Figure 3b. Please add what 3,5 and 7 mean.

We have added this information in the new Figure 2 legend.

Response to referee #3:

Thank you for your time reading and providing comments on our manuscript. Please see below our response to your comments.

Altogether, I think this is a fantastic manuscript and I only have minor requests of the authors.

Thank you!

1. In figure 2 legend, can you write out what I3, I4E, I4L...etc, mean to make it easier for the reader?

We have added explanations of the stages in the new Figure 1 legend.

2. It would be nice to see an ontogenetic series of the wings from the 3rd nymphal instar to adulthood in 12:12, 18:6, and perhaps the photoperiod switching experiments. Specifically, it would be nice if you can take the wings off the animal, flatten under a microscope slide, and image to scale. I ask this because it seems that instars 3 and 4 are key stages where the growth might stunt, and looking that *Micropterus* wings, it is almost as if growth is stunted but differentiation of the anterior-posterior and proximo-distal axis is not. It would be nice to have a visual representation of this for the readers.

As suggested we have added images of wing bud progression from Instar 3 – Instar 5. We chose to display it un-dissected since the wing buds looked very de-formed/compressed when put under a microscope slide.

3. This is just a thoughtful note for the authors. I am not requesting any of the following be included in the manuscript. It seems that the short-wing morphs are left with a fully organized, yet vestigial wing. It reminds me of the difference in the forewing and hindwing in Diptera, wherein the hindwing has been reduced to become the haltere. This essentially happens by dampening the growth regulatory effects of virtually all of the morphogens (Pavlopoulos & Akam, 2011; Weatherbee et al., 1998). In normal wing development, the response to morphogens seems to be linked to the temporal transcriptional turnover directed by rising ecdysone levels (Mirth et al., 2009; Oliveira et al., 2014), and photoperiod has been associated with temporal alterations in ecdysone biosynthesis in many polyphenisms (Nijhout, 1999). I wonder whether this wing polyphenism is determined by affecting ecdysone biosynthesis, and the wings have

evolved multiple thresholds for ecdysone, wherein crossing the minimum threshold facilitates the progression of AP and PD patterning, and crossing the next threshold facilitates proliferation and this might be realizable by toying around with the crosstalk between ecdysone signaling and Hippo/Warts signaling (McKenna et al., 2019). This last reference discusses the literature of ecdysone-Hippo/Warts cross talk.

We are very grateful for these thoughts and we are currently in the process of repeating Ecdysone/Methoprene hormone treatments from some earlier experiments in our different photoperiod treatment to see if this has a photoperiod dependent effect.

Appendix B

UPPSALA
UNIVERSITET

Uppsala 04.03.2022

Dear editor,

Please find resubmission for manuscript RSPB-2021-2764 entitled: “Photoperiod controls wing polyphenism in a water strider independently of insulin receptor signaling” for potential publication in PRSB attached.

We are very grateful to the two reviewers and the associate editor for their helpful comments and thoughts on this version of the manuscript and we have now revised it according to the suggested changes.

More specifically we have:

- 1) Corrected the typos highlighted by the associate editor and the two reviewers.
- 2) Clarified the methods and results to better describe the experiments and the results from them.
- 3) Changed the text according to many of the suggestions from reviewer 2 to make the reading more streamlined and the description of photoperiods explicit.

Detailed responses to editor and reviewers’ comments can be found below with our response below the reviewer comments.

Please do not hesitate to contact us if you have any questions and we look forward to hear from you.

Best wishes,

Erik Gudmunds (on behalf of all authors)

Responses to referee comments for the manuscript “Photoperiod controls wing polyphenism in a water strider independently of insulin receptor signaling components”.

We have put comments from editor and referees in bold text and our responses in normal font. Changes in the manuscript are written in italic font.

Associate Editor

Please also correct the following typos and check the manuscript for similar typos throughout. These line numbers correspond to the ‘track changes’ version

Thank you for pointing these out, we have corrected all the listed typos.

Response to referee 2:

Thank you for the comments and suggestions, see below for the changes we have made.

Abstract lines 28-30. This study failed to find genetic pathways underlying wing development. Ultimately, the end of the abstract needs some wider take home message.

We changed the last sentence in the abstract to include a point made in the conclusion about our data suggestion there are multiple genetic origins to wing polyphenism in insects.

L29-31: *“Our results indicate that the multitude of possible cues that trigger wing polyphenism can be mediated through different genetic pathways and that there are multiple genetic origins to wing polyphenism in insects.”*

Line 149. a binomial error structure and logit link function

We have changed the description of the statistical model to what you suggest.

Results: Chi-square values should be quoted to two decimal places.

We have changed all chi-square values to have two decimal places.

line 344. Please summarize your experiments more clearly.

We have changed the sentence to be explicit about what was done and which results were obtained:

L363-365. *“Variation in nutrition had no effect on wing morph induction but increasing density was associated with an increase of macropterous morphs.”*

Lines 354-358. Please add a reference.

Thank you for pointing this out. We have now added the relevant reference.

Response to referee 1:

Thank you for the time and effort you've invested into the manuscript and for the comments and suggestions on how to improve it. We have made many changes the text based on your suggestions made directly in the text, and note that they improved the clarity and flow of the text. Please see below our response to the specific comments.

1. (Methods) How many bugs and how many times were stocks 'replenished', even roughly?

We have checked with the lab who we got the population from and it turns out that it was only replenished once with 30 individuals. We have changed the text accordingly:

L87-89. *"The G. buenoi population was originally collected from a pond in Toronto, Ontario, Canada, during 2012, and was replenished in 2015 with 30 individuals from a population in the same area."*

2. (Methods) I assume you actually mean forewings or hemelytra, or did you actually tease out the reduced flight wings?

We mean forewings and have clarified that at relevant places.

3. (Methods) Densities may have varied box to box during the experiment to assess density effects. I guess that the 'numbers out' provide some rough and ready assessment of how important this could be. It might be important to assure readers that the output numbers for the 4 density treatments were statistically different and distributed on the gradient of abundance as expected.

We have provided the ending density in terms of mean relative density of replicates in each treatment.

L176-179. *"The mean relative densities at the end of the experiment were 1.6, 3.4, 5.5, 9.4 individuals per 100 cm² for 18L:6D and 1.9, 3.4, 6.2 11.4 individuals per 100 cm² for 12L:12D for the low, medium, high and extreme density treatments, respectively."*

4. (Methods) Under the Nutrition experiments and elsewhere I think that the authors must be explicit about photoperiod in these experiments, if it is the same as the hatching photoperiod, they could write more simply something like "groups ... were hatched and reared in both photoperiods". These kinds of details are necessary to meet the criterion of repeatability. The way I read it right now is that starting groups of treatments were a random mix of first instars hatched at the two different photoperiods, and thus the experiments are flawed by this uncontrolled factor, although I imagine that this is not the case. Perhaps it is thought that hatching photoperiod does not matter, but if so this needs to be clear. This is also a good place to mention, that I think that somewhere toward the beginning of the paper the authors must explain what they are calling 'long' and 'short' photoperiods and be sure to adopt a consistent terminology and designation throughout the paper. I personally prefer the simple designations like, for example, '16L:8D' because they are unaambiguous, but I think that anything will work as long as it is defined and consistently employed. Also, I think that the relationship between constant photoperiods administered in the lab and natural daylight under which the

polyphenism has evolved and is regulated in nature is much better rationalized in this version of the manuscript; however, I would look carefully at this aspect throughout the MS, say, e.g., in Sect. 3.a., to see if further improvement is possible.

We have added a sentence to clearly spell out that the photoperiod for hatching and for rearing during the nutrition regimes was the same for all individuals in the experiment. L192-193. *“For each individual in the experiment photoperiod was kept the same as hatching photoperiod (12L:12D or 18L:6D) throughout development.”*

Also, we have changed the terminology for photoperiod descriptions, so that we now use e.g. 16L:8D and not 16:8 L:D or e.g. only “long”. We agree that it is better to use an unambiguous terminology. In some instances, we write “long” but put the actual photoperiod in parenthesis, e.g. L359-361:

“Furthermore, we showed that a shift from short (12L:12D) to a long (18L:6D) photoperiod is highly inductive of the micropterous morph in a developmental window variably expressed from at least instar 3 until at latest two days into instar 5.”

5. (Methods) Although things are much improved in this version, I still struggle a little with understanding exactly what was done on the molecular side. It would be useful for readers like me, if the authors could succinctly explain what they were aiming to achieve with the injections. Expressions like ‘knockdown’ or ‘depletion’ give some notion that the objective was to nullify the action of particular genes, but can one clearly explain the mechanics of how this is being accomplished. There are a series of comments on the MS in track changes to suggest where more complete discussion could help neophyte readers better understand what was done

Unfortunately, we are too limited in space to describe the mechanisms of RNAi. But we have added a sentence to clearly state what the dsGFP treatment controls for:

L234-238. *“The negative control in the RNAi experiments is the dsGFP treatment, where dsRNA with the sequence of green fluorescent protein is injected into individuals but should not interfere with gene expression, thus controlling for potential effects of the injection procedure in itself on wing morph determination.”*

6. (Methods) The last sentence of the Methods section is quite uninterpretable to me. For me to understand, more context is required.

We have altered the sentence to make clear that the analysis has to do with estimating sequence similarities between the *inr* gene coding sequences:

L247-249. *“The estimation of pairwise sequence identity of the *inr* genes coding sequences was done with Muscle (version 3.8.425 (alignment in Geneious (version 2021.1.1) using standard parameters.”*

7. (Results) I interpret Fig. 2A in a way somewhat different from what is written at this point. Please consider the rewording that I’ve suggested or clarify what is meant.

We have made changes on the description of the density experiment results in the way you suggested in the text.

L285-291. *“Almost all nymphs reared in 12L:12D across all density regimes became macropterous (3 out of 574 in total became micropterous) whereas in 18L:6D most individuals became micropterous across all densities (Figure 2A). The frequency of the macropterous morph in 18L:6D significantly increased with density ($\chi^2 = 24,39$, d.f. = 3, $P = 2 \times 10^{-5}$). Interestingly, the frequency of macropterous individuals in 18L:6D in the extreme density was higher among females than males ($\chi^2 = 18,23$, d.f. = 6, $P = 0.006$).”*

8. (Results) It took me a little while to sort out the meaning of the text here. I suppose you mean to refer only to the first instars reared at 18L:D6, given the df associated with the test. Given the low sample for those reared with low diets, I am not sure that a χ^2 is valid, and as you say the results must be treated cautiously.

Yes, we have clarified that we mean the individuals for which the nutrition regimes started in the first instar:

L294- 297. *“While we found a significant difference in the frequency of mesopterous adults among individuals treated with different nutrient regimes from instar 1 (Figure 2B, $\chi^2 = 17,08$, d.f. = 2, $P = 0.0002$), sample size was very low in the low diet and thus this result should be treated with caution.”*

9. (Methods) Were these experiments controlled for the possible simple effects of injection? It seems to me that injection of equal amounts of saline or some other innocuous solution is required to be sure of interpretations.

Yes, see above.

10. (Discussion) I agree that this is one way to interpret the results. I prefer the idea that there is individual variation in any population for critical stage, so there is always room for selection to move a population with changing climate. In some years, an individual that commits in 3rd instar may lose some or all of it's reproductive output the because there is an early winter. However, as winters drift later, such individuals would be favoured. I'd bet that individuals have relatively narrow windows for induction although the window may be wide in a population. I may be wrong, but I don't think one can prove the point with what we know and I think the discussion ought to be equivocal about it.

We have specified that the window is for the population and that the results suggests genetic variation in sensitive stage for induction.

L432-434. *“For our population, exposure to a long photoperiod (18L:6D) was highly inductive of the micropterous morph and that the sensitive window for this induction lasts from at least instar 3 until latest day 2 in instar 5.”*

L444-447. *“Interestingly, there is variation in wing morphs within some experimental groups (e.g. I5E and I5M in the 12L:12D to 18L:6D shift, Figure 1E), which suggests genetic variation in the sensitive stage for induction.”*

11. (Discussion) How do we know that even this qualified statement is correct? Wing buds begin to appear in instar 3 and are obvious in instar 4. Wings appear at the adult molt and as far as I could see, there is no difference between wing bud length between nymphs that become either micropters or macropers.

As you say, there is no readily detectable difference in instar 5 wings buds between LW or SW destined individuals. Meaning that whatever size differences we see in the adult forewings and hindwings must come from differential growth in instar 5. That said, the molecular/physiological system that governs the growth can, as our data shows, be active as early as instar 3.